# FRAMEBRIDGE: IMPROVING IMAGE-TO-VIDEO GENERATION WITH BRIDGE MODELS

## ABSTRACT

Image-to-video (I2V) generation is gaining increasing attention with its wide application in video synthesis. Recently, diffusion-based I2V models have achieved remarkable progress given their novel design on network architecture, cascaded framework, and motion representation. However, restricted by their noise-to-data generation process, diffusion-based methods inevitably suffer the difficulty to generate video samples with both appearance consistency and temporal coherence from an uninformative Gaussian noise, which may limit their synthesis quality. In this work, we present FrameBridge, taking the given static image as the prior of video target and establishing a tractable bridge model between them. By formulating I2V synthesis as a *frames-to-frames* generation task and modeling it with a *data-to-data* process, we fully exploit the information in input image and facilitate the generative model to learn the image animation process. In two popular settings of training I2V models, namely fine-tuning a pre-trained text-to-video (T2V) model or training from scratch, we further propose two techniques, SNR-Aligned Fine-tuning (SAF) and neural prior, which improve the fine-tuning efficiency of diffusion-based T2V models to FrameBridge and the synthesis quality of bridge-based I2V models respectively. Experiments conducted on WebVid-2M and UCF-101 demonstrate that: (1) our FrameBridge achieves superior I2V quality in comparison with the diffusion counterpart (zero-shot FVD 95 vs. 192 on MSR-VTT and non-zero-shot FVD 122 vs. 171 on UCF-101); (2) our proposed SAF and neural prior effectively enhance the ability of bridge-based I2V models in the scenarios of fine-tuning and training from scratch. Demo samples can be visited at: `https://framebridgei2v.github.io/`.

## 1 INTRODUCTION

Image-to-video (I2V) generation, commonly referred as image animation, aims at generating consecutive video frames from a static image (Xing et al., 2023; Ni et al., 2023; Zhang et al., 2024a; Guo et al., 2023; Hu et al., 2022), *i.e.*, a *frame-to-frames* generation task where maintaining appearance consistency and ensuring temporal coherence of generated video frames are key evaluation criteria (Xing et al., 2023; Zhang et al., 2024a). With the recent progress in video synthesis (Brooks et al., 2024; Yang et al., 2024; Blattmann et al., 2023; Bao et al., 2024), several diffusion-based I2V frameworks have been proposed, with novel designs on network architecture (Xing et al., 2023; Zhang et al., 2024a; Chen et al., 2023b; Ren et al., 2024; Lu et al., 2023), cascaded framework (Jain et al., 2024; Zhang et al., 2023), and motion representation (Zhang et al., 2024b; Ni et al., 2023). However, although these methods have demonstrated the potential of diffusion models (Ho et al., 2020; Song et al., 2020) in I2V synthesis, restricted by their *noise-to-data* generation process, they inevitably suffer the difficulty to generate video samples required by both appearance consistency and temporal coherence from uninformative random noise. With the *noise-to-data* sampling trajectory which is inherently mismatched with the *frame-to-frames* synthesis process of I2V task, previous diffusion-based methods increase the burden of generative models, which may result in limited synthesis quality.

In this work, we present FrameBridge, a novel I2V framework to model the *frame-to-frames* synthesis process with recently proposed *data-to-data* generative framework (Chen et al.; Liu et al., 2023; Chen et al., 2023c). Specifically, given the input image and video target, we first leverage variational auto-encoder (VAE) based compression network to transform them into continuous latent representations, and then take their latent representations as boundary distributions, *i.e.*, prior and target, to establish our *data-to-data* generative framework. Considering the static image has already been an informative prior for each of the consecutive frames in video target, we naturally replicate it to obtain the prior of the whole video clip, constructing the *frames-to-frames* pairs for the prior-to-target generation process in FrameBridge. Standing on constructed frames-to-frames pairs, we establish bridge models (Tong et al., 2023; Zhou

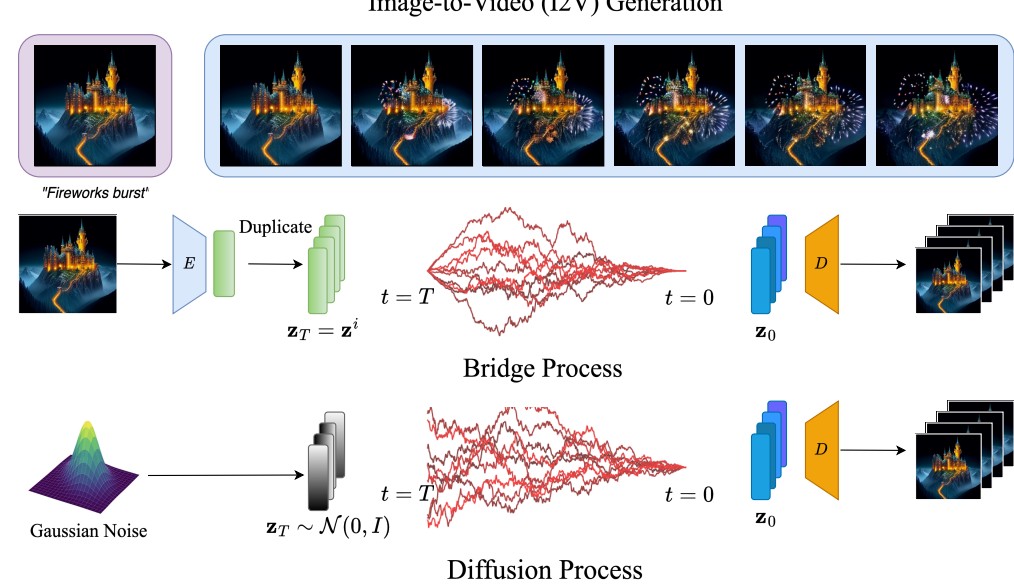

Figure 1: **Overview of FrameBridge and diffusion-based I2V models**. The sampling process of FrameBridge (upper) starts from a deterministic data point, while diffusion models (lower) synthesize videos from Gaussian noise.

et al., 2023; Chen et al., 2023c) between them to learn the I2V synthesis with Stochastic Differential Equation (SDE) based generation process, implicitly modeling the image animation with SDE-based generative models. In comparison with previous diffusion works, our FrameBridge utilizes given static image as the prior of video target, which is advantageous on preserving the appearance details of input image than conditionally generating video samples from random noise. Moreover, our frames-to-frames bridge model naturally learns image animation in model training rather than learning the image-conditioned noise-to-video generation. The improved consistency between generative framework and I2V task, *i.e.*, *data-to-data* for *frame-to-frames*, tends to benefit temporal coherence for I2V synthesis.

In practice, I2V systems usually leverage the potential of a pre-trained diffusion-based text-to-video (T2V) model (Xing et al., 2023; Chen et al., 2023b; Ma et al., 2024a) with a fine-tuning process, to reduce the requirements of image-video data pairs and the computational resources at training stage. Therefore, in the application of FrameBridge, we aim to efficiently fine-tune a pre-trained diffusion-based T2V model to our proposed bridge-based I2V model. Toward this target, we propose SNR-Aligned Fine-tuning (SAF), which aligns the two frameworks, *i.e.*, diffusion models and bridge models, by adjusting the bridge process. Specifically, we first reparameterize the bridge process in FrameBridge, enabling the noisy intermediate latents of our frames-to-frames process to be aligned with the ones in the noise-to-data process of pre-trained diffusion models. Then, we change the timestep to match the signal-to-noise (SNR) ratio between the input of the bridge model and the pre-trained diffusion model. SAF avoids the mismatch between the two different generative frameworks at pre-training and fine-tuning stage, and therefore improves the synthesis quality of FrameBridge when adapting pre-trained T2V diffusion models.

Compared to diffusion models using Gaussian prior, FrameBridge takes static image as prior to improve I2V performance. A natural question to our proposed method is if we could further improve the I2V quality with a stronger prior. We answer this question by further exploiting the prior of our method. Given a static image, we design a one-step mapping-based network and optimize it with the video target, extracting a *neural prior* from the image for the video target. Compared to input image, this neural prior reduces the distance between prior and video target to a greater extent, and alleviates the burden of generative models further. Although more advanced methods can be leveraged to extract more informative neural prior, we empirically find that a coarse estimation for video target at the cost of a single sampling step has already been beneficial to FrameBridge. This further verifies our motivation to present FrameBridge and shows a novel method to enhance bridge-based I2V models. In this work, our contributions can be summarized as follows:

- We propose FrameBridge, making the first attempt to model the frame-to-frames generation task of I2V with a data-to-data generative framework.

- We present two novel techniques, SAF and neural prior, further improving the performance of FrameBridge when fine-tuning from pre-trained T2V diffusion models and training from scratch respectively.

- We conduct experiments on two I2V benchmarks by training FrameBridge on WebVid-2M (Bain et al., 2021) and UCF-101 (Soomro, 2012). FrameBridge fine-tuned with SAF reduces the zero-shot FVD (Unterthiner et al. (2018); lower is better) from 176 to 83 on MSR-VTT (Xu et al., 2016), and FrameBridge with neural prior trained from scratch reduces the non-zero-shot FVD from 171 to 122 on UCF-101, highlighting the superiority of FrameBridge to their diffusion counterparts and the effectiveness of SAF and neural prior.

## 2 RELATED WORKS

**Diffusion-based I2V Generation** Diffusion models have recently achieved remarkable progress in I2V synthesis (Blattmann et al., 2023; Chen et al., 2023a; Li et al., 2024). Previous works have explored multi-stage generation system (Jain et al., 2024; Zhang et al., 2023; Shi et al., 2024) fusion module (Wang et al., 2024; Ren et al., 2024) and improved network architectures (Wang et al., 2024; Xing et al., 2023; Ma et al., 2024a; Chen et al., 2023b; Ren et al., 2024), while their generative framework remains the *noise-to-data* one of diffusion models, which may be inefficient for I2V synthesis. To improve the uninformative prior, PYoCo (Ge et al., 2023) recently proposes to use correlated noise for each frame in both training and inference. ConsistI2V (Ren et al., 2024), FreeInit (Wu et al., 2023), and CIL (Zhao et al., 2024) present training-free strategies to better align the training and inference distribution of diffusion prior, which is popular in diffusion models (Lin et al., 2024; Podell et al., 2023; Blattmann et al., 2023; Girdhar et al., 2023). These strategies focus on improving the noise distribution to enhance the quality of synthesized videos, while they still suffer the restriction of noise-to-data diffusion framework, which may limit their endeavor to utilize the entire information (*e.g.*, both large-scale features and fine-grained details) contained in the given image. In this work, we propose a *data-to-data* framework and utilize deterministic prior rather than Gaussian noise, allowing us to leverage the clean input image as prior information.

**Bridge Models** Recently, bridge models (Chen et al.; Tong et al., 2023; Liu et al., 2023; Zhou et al., 2023; Chen et al., 2023c), which overcome the restriction of Gaussian prior in diffusion models, have gained increasing attention. They have demonstrated the advantages of *data-to-data* generation process over the *noise-to-data* one on image-to-image translation (Liu et al., 2023; Zhou et al., 2023) and text-to-speech synthesis (Chen et al., 2023c) tasks. In this work, we make the first attempt to extend bridge models to I2V synthesis and further propose two improving techniques for bridge models, enabling efficient fine-tuning from diffusion models and stronger prior for video target.

## 3 BACKGROUND

**Problem Formulation** I2V aims at generating an video clip $\mathbf{v} \in \mathbb{R}^{L \times H \times W \times 3}$ with a number of $L$ frames from a static image, *e.g.*, the initial frame $v^i \in \mathbb{R}^{H \times W \times 3}$ of video clip $\mathbf{v}$. In I2V systems (Xing et al., 2023; Blattmann et al., 2023), an VAE-based compression network is usually leveraged to first transform the video $\mathbf{v}$ into a latent $\mathbf{z} \in \mathbb{R}^{L \times h \times w \times d}$ in a per-frame manner with a pre-trained image encoder $\mathcal{E}(\mathbf{v})$, where $h = \frac{H}{p}$, $w = \frac{W}{p}$, $p > 1$ and $d$ are the spatial compression ratio and the number of output channels. Then, generative models are designed in this compressed space to learn the conditional distribution $p_{\mathbf{z}}(\mathbf{z}|z^i, c)$, where $z^i \in \mathbb{R}^{h \times w \times d}$ is the compressed latent of the initial frame $v^i$ and $c$ denotes other guidance such as the text prompt (Ma et al., 2024a; Chen et al., 2023b) or the image class condition (Ni et al., 2023; Zhang et al., 2024b). In sampling, we first synthesize the latent $\mathbf{z}$ conditioned on the latent of given static frame $z^i$, and then decode the video clip with pre-trained VAE decoder $\mathcal{D}(\mathbf{z})$. As formulated, I2V synthesis actually seeks to generate consecutive frames in $\mathbf{v}$ from a single frame $v^i$, *i.e.*, a *frame-to-frames* generation process where the appearance details of $v^i$ should be preserved in generated $\mathbf{v}$ and the animation in $\mathbf{v}$ should start from $v^i$ and remain temporal coherent.

**Diffusion-based I2V Synthesis** Diffusion models (Ho et al., 2020; Sohl-Dickstein et al., 2015) have been popularly leveraged to learn the conditional distribution $p_{\mathbf{z}}(\mathbf{z}|z^i, c)$. These models are composed of two processes. A forward process gradually converts the video latent $p_0(\mathbf{z}_0|z^i, c) \triangleq p_{data}(\mathbf{z}_0|z^i, c)$ to a known prior distribution $p_{T,diff}(\mathbf{z}_T) \triangleq$

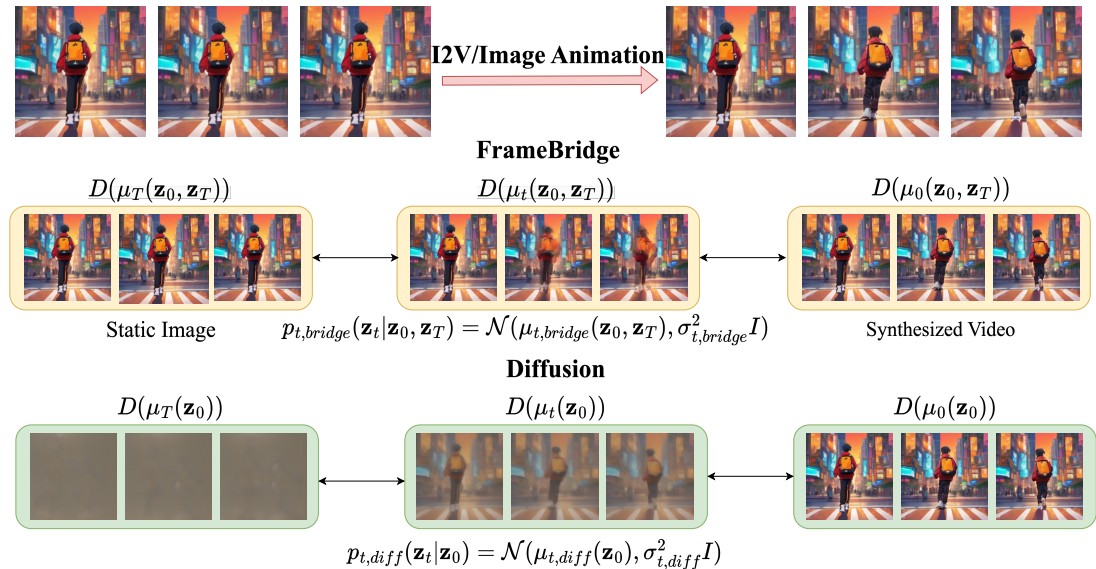

Figure 2: **Visualization for the mean value of marginal distributions**. We visualize the decoded mean value of bridge process and diffusion process. The prior and target of FrameBridge are naturally suitable for I2V synthesis.

$p_{prior,diff}(\mathbf{z}_T)$ with a forward SDE (Song et al., 2020):

$$d\mathbf{z}_t = \boldsymbol{f}(t)\mathbf{z}_t dt + g(t)d\mathbf{w}, \quad \mathbf{z}_0 \sim p_{data}(\mathbf{z}_0|z^i, c), \tag{1}$$

where $\mathbf{w}$ is a Wiener process, $\boldsymbol{f} : \mathbb{R}^D \times [0,T] \rightarrow \mathbb{R}^D$ is the drift coefficient, and $g : [0,T] \rightarrow \mathbb{R}$ is the diffusion coefficient. $D$ represents the dimension of data $\mathbf{z}_0$. The marginal distribution of $\mathbf{z}_t$ satisfies $p_{t,diff}(\mathbf{z}_t|z^i, c) = \int_{\mathbf{z}_0 \sim p_0(\mathbf{z}_0|z^i, c)} p_{t,diff}(\mathbf{z}_t|\mathbf{z}_0) p_0(\mathbf{z}_0|z^i, c) d\mathbf{z}_0$. Here $p_{t,diff}(\mathbf{z}_t|\mathbf{z}_0) = \mathcal{N}(\alpha_t \mathbf{z}_0, \sigma_t^2 I), \alpha_t = e^{\int_0^t f(\tau)d\tau}, \sigma_t^2 = \alpha_t^2 \int_0^t \frac{g(\tau)^2}{\alpha_\tau^2} d\tau$ (Kingma et al., 2021).

Given the forward process defined by eq. (1), there exists a reverse process with a backward SDE which shares the same marginal distribution $p_{t,diff}(\mathbf{z}_t|z^i, c)$ (Song et al., 2020):

$$d\mathbf{z}_t = \left[\boldsymbol{f}(t)\mathbf{z}_t - g(t)^2 \nabla_{\mathbf{z}_t} \log p_{t,diff}(\mathbf{z}_t|z^i, c)\right] dt + g(t)d\bar{\mathbf{w}}, \quad \mathbf{z}_T \sim p_{prior,diff}(\mathbf{z}_T), \tag{2}$$

where $\bar{\mathbf{w}}$ is a backward Wiener process.

To learn the unknown term in eq. (2), *i.e.*, the score function $\nabla_{\mathbf{z}_t} \log p_{t,diff}(\mathbf{z}_t|z^i, c)$, usually a U-Net (Ronneberger et al., 2015; Ho et al., 2020) or DiT (Peebles & Xie, 2023; Bao et al., 2023) based neural network is optimized with a denoising objective to predict the noise:

$$\mathcal{L}_{diff}(\theta) = \mathbb{E}_{(\mathbf{z}_0,z^i,c)\sim p_{data}(\mathbf{z}_0,z^i,c),t\sim q(t),\mathbf{z}_t\sim p_{t,diff}(\mathbf{z}_t|\mathbf{z}_0)} \left[\lambda(t) \left\|\boldsymbol{\epsilon}_\theta(\mathbf{z}_t,t,z^i,c) - \frac{\mathbf{z}_t - \alpha_t \mathbf{z}_0}{\sigma_t}\right\|^2\right], \tag{3}$$

where $q(t)$ is a distribution of $t$ supporting on $[0,T]$, $\lambda(t)$ is a time-dependent weight function, and $\boldsymbol{\epsilon}_\theta(\mathbf{z}_t, t)$ is an alternative parameterization method of the score function (Ho et al., 2020).

**Limitations** As shown, the forward process of diffusion models gradually injects noise into data samples, which results in a boundary distribution at $t = T$ sharing the same distribution with the injected noise, *e.g.*, the standard Gaussian noise $\boldsymbol{\epsilon} \sim \mathcal{N}(\mathbf{0}, \boldsymbol{I})$. Therefore, in generation, their sampling process has to start from the uninformative prior distribution $p_{prior,diff}(\mathbf{z}_T) \sim \mathcal{N}(\mathbf{0}, \boldsymbol{I})$ and then iteratively synthesize the video latent $\mathbf{z}_0$ with learned conditional score function $\nabla_{\mathbf{z}_t} \log p_t(\mathbf{z}_t|z^i, c)$. Formulating a *frame-to-frames* generation task with a conditional *noise-to-data* sampling process, diffusion-based I2V systems suffer the difficulties to generate high-quality samples from uninformative Gaussian noise while preserving the appearance details with condition and keeping the temporal coherence, which may result in limited I2V performance.

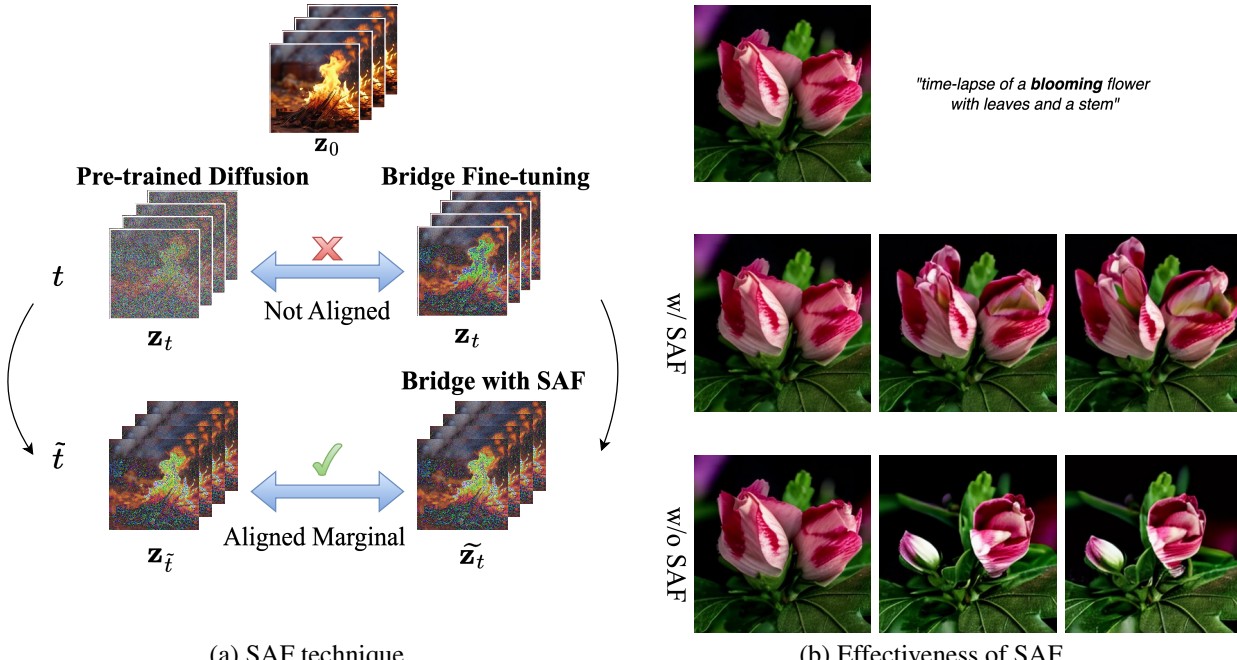

(a) SAF technique

(b) Effectiveness of SAF

Figure 3: **SNR-Aligned Fine-tuning for FrameBridge.** (a) SAF technique aligns the marginal distributions of Frame-Bridge with that of pre-trained Gaussian diffusion models. (b) FrameBridge with SAF can better leverage the capability of pre-trained models.

## 4 METHOD

### 4.1 MOTIVATION

As discussed above, on one hand, the static image in I2V generation has already provided strong condition information, *e.g.*, the appearance details which should be preserved in generated video. However, the prior of diffusion models ignore this information, rendering diffusion-based I2V systems regenerate entire video information from Gaussian noise. On the other hand, the image animation process should be temporal coherent, ensuring the consistency between video frames. However, the prior of diffusion models makes previous works generate each frame from random noise. Although the generation is conditioned on static image, it would be difficult for a *noise-to-data* generation process to guarantee the temporal coherence between generated frames.

Toward reducing the mismatch between the generation process and the I2V task, we make the first attempt to establish *data-to-data* process, *i.e.*, bridge models (Chen et al., 2023c; Zhou et al., 2023), for the *frame-to-frames* synthesis of I2V. Taking the static image as the prior of consecutive frames in video, we build a bridge between the initial frame and the following ones. In generation, each video frame is generated from the initial frame, which is naturally helpful to preserve appearance details in I2V synthesis. Moreover, as our model learns the *frame-to-frames* synthesis instead of conditional *noise-to-frames* generation, we facilitate our model to focus on learning image animation, which benefits temporal coherence of generated video.

### 4.2 FRAMEBRIDGE

Considering the given image, *i.e.*, initial frame $z^i$, has provided the appearance details and the starting point of animation for video target, we take it as the prior of following frames. To construct the boundary distributions for bridge models, we replicate the image latent $z^i$ for $L$ times along temporal axis to obtain $\mathbf{z}^i \in \mathbb{R}^{L \times h \times w \times d}$ as the prior of video latent $\mathbf{z} \in \mathbb{R}^{L \times h \times w \times d}$, and establish the bridge process as follows.

**Bridge Process**  In Figure 1, we present the overview of FrameBridge and compare it with diffusion-based I2V generation. Different from diffusion-based I2V models using uninformative Gaussian prior, our FrameBridge replaces the Gaussian prior with a Dirac prior $\delta_{\mathbf{z}^i}$, building a bridge process (Zhou et al., 2023) to connect the video target and the replicated image prior $p_{prior,bridge}(\mathbf{z}_T|i,c) \triangleq \delta_{\mathbf{z}^i}(\mathbf{z}_T)$. Specifically, the forward process is changed from eq. (1) in diffusion models to:

$$\mathrm{d}\mathbf{z}_t = \left[\boldsymbol{f}(t)\mathbf{z}_t + g(t)^2\boldsymbol{h}(\mathbf{z}_t,t,\mathbf{z}_T,z^i,c)\right]\mathrm{d}t + g(t)\mathrm{d}\boldsymbol{w}, \quad \mathbf{z}_0 \sim p_{data}(\mathbf{z}_0|z^i,c), \quad \mathbf{z}_T = \mathbf{z}^i, \tag{4}$$

where $\boldsymbol{h}(\mathbf{z}_t,t,\mathbf{z}_T,z^i,c) \triangleq \nabla_{\mathbf{z}_t}\log p_{T,diff}(\mathbf{z}_T|\mathbf{z}_t)$ and $p_{T,diff}(\mathbf{z}_T|\mathbf{z}_t)$ is the marginal distribution of diffusion process shown in eq. (1). For bridge process, we denote the marginal distribution of eq. (4) as $p_{t,bridge}(\mathbf{z}_t|z^i,c)$. Similar to the forward SDE eq. (1) in diffusion process, the forward process of bridge models eq. (4) also has a reverse process, which shares the same marginal distribution $p_{t,bridge}(\mathbf{z}_t|z^i,c)$ and can be represented by the backward SDE:

$$\mathrm{d}\mathbf{z}_t = \left[\boldsymbol{f}(t)\mathbf{z}_t - g(t)^2(\boldsymbol{s}(\mathbf{z}_t,t,\mathbf{z}_T,z^i,c) - \boldsymbol{h}(\mathbf{z}_t,t,\mathbf{z}_T,z^i,c))\right]\mathrm{d}t + g(t)\mathrm{d}\bar{\boldsymbol{w}}, \quad \mathbf{z}_T = \mathbf{z}^i, \tag{5}$$

where $\boldsymbol{s}(\mathbf{z}_t,t,\mathbf{z}_T,z^i,c) \triangleq \nabla_{\mathbf{z}_t}\log p_{t,bridge}(\mathbf{z}_t|\mathbf{z}_T,z^i,c)$. The change from the diffusion to the bridge process removes the restriction of noisy prior, allowing the generation process to start from a static image rather than previous Gaussian noise. Moreover, as the perturbation kernel $p_{t,bridge}(\mathbf{z}_t|\mathbf{z}_0,\mathbf{z}_T,z^i,c)$ in bridge process remains Gaussian (Appendix A), it facilitates us to find connections between the marginal distribution, *i.e.*, the intermediate representations of diffusion and bridge process, and then leverage the power of pre-trained diffusion models for bridge models.

**Training Objective**  Analogous to diffusion models, we use a SDE solver to solve eq. (5) when sampling videos. Since $\boldsymbol{h}(\mathbf{z}_t,t,\mathbf{z}_T,z^i,c)$ can be calculated analytically (see Appendix A), we only need to estimate the unknown term $\boldsymbol{s}(\mathbf{z}_t,t,\mathbf{z}_T,z^i,c)$ with neural networks (Kingma et al., 2021). After parameterization (more details can be found in Appendix A), we train FrameBridge models $\boldsymbol{\epsilon}_\theta^{\hat{\Psi}}(\mathbf{z}_t,t,\mathbf{z}_T,z^i,c)$ with the denoising objective (Chen et al., 2023c):

$$\mathcal{L}_{bridge}(\theta) = \mathbb{E}_{\substack{(\mathbf{z}_0,z^i,c)\sim p_{data}(\mathbf{z}_0,z^i,c),\\ \mathbf{z}_T=\mathbf{z}^i, t, \mathbf{z}_t\sim p_{t,bridge}(\mathbf{z}_t|\mathbf{z}_0,\mathbf{z}_T,z^i,c)}}\left[\left\|\boldsymbol{\epsilon}_\theta^{\hat{\Psi}}(\mathbf{z}_t,t,\mathbf{z}_T,z^i,c) - \frac{\mathbf{z}_t - \alpha_t\mathbf{z}_0}{\sigma_t}\right\|^2\right]. \tag{6}$$

The training of FrameBridge resembles that of Gaussian diffusion-based I2V models: We first sample a video latent $\mathbf{z}_0$ and the condition $c$ from training set, extracting the first frame of $\mathbf{z}_0$ to construct $\mathbf{z}^i$. The primary difference lies in the Gaussian perturbation kernel $p_{t,bridge}(\mathbf{z}_t|\mathbf{z}_0,\mathbf{z}_T,z^i,c)$ of eq. (6). As we replace the Gaussian prior with a deterministic data point $\mathbf{z}_T$, the mean value is an interpolation between data and $\mathbf{z}_T$ instead of the decaying data in diffusion models, naturally preserving more data information and facilitating generative models to learn image animation rather than regenerating the information provided in static image.

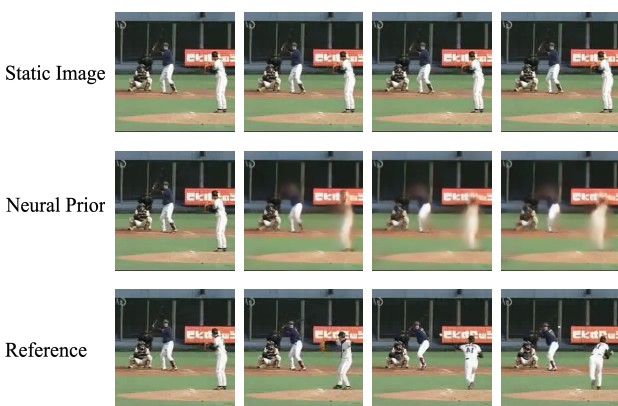

Static Image

Neural Prior

Reference

Figure 4: **Case of neural prior.** The neural prior provides more motion information than a static image, and is semantically closer to the reference video in dataset.

**Bridge Process vs Diffusion Process**  To demonstrate the advantages of bridge process in I2V synthesis, we visualize the data part, *i.e.*, the mean function of bridge and diffusion process, in Figure 2. As shown, when replicating the initial frame, I2V synthesis can be formulated as a *frames-to-frames* generation task. With the *data-to-data* bridge process, the boundary distributions of our FrameBridge, *i.e.*, the prior and the target, have been an ideal fit for the I2V task, which is helpful for generative models to focus on modeling the image animation process. In the meanwhile, as seen from our intermediate representations, the data information, *e.g.*, appearance details, is well preserved during the bridge process. In comparison, the prior and intermediate representations of diffusion process contain rare or coarse information of the target, which is uninformative and requires diffusion models to generate entire video information from scratch conditioned on static image.

Table 1: Zero-shot I2V generation on UCF-101 and MSR-VTT ($256 \times 256$, 16 frames). w/o SAF means FrameBridge without SAF techniques when fine-tuning. Results marked by $*$ are reported in Xing et al. (2023). Some metrics are not applicable for SVD as the I2V model of SVD generate videos with 14 frames and other models in the table generate videos with 16 frames. For each metric, we mark the best one with † and the second one with ‡.

| Method | UCF-101 | | | MSR-VTT | | |
|---|---|---|---|---|---|---|
| | FVD ↓ | IS ↑ | PIC ↑ | FVD ↓ | CLIPSIM ↑ | PIC ↑ |
| DynamiCrafter (Xing et al., 2023) | 485 | 29.46 | 0.6266 | 192 | 0.2245 | 0.6131 |
| DynamiCrafter$^*$ | 429 | – | 0.6078 | 234 | – | 0.5803 |
| I2VGen-XL$^*$ (Zhang et al., 2023) | 571 | – | 0.5313 | 289 | – | 0.5352 |
| SVD (Blattmann et al., 2023) | **235**$^‡$ | – | – | 114 | – | – |
| SEINE (Chen et al., 2023b) | 461 | 22.32 | 0.6665 | 245 | **0.2250**$^†$ | 0.6848 |
| ConsistI2V (Ren et al., 2024) | **202**$^†$ | 39.76 | **0.7638**$^†$ | 106 | 0.2249 | **0.7551**$^†$ |
| SparseCtrl (Guo et al., 2025) | 722 | 19.45 | 0.4818 | 311 | 0.2245 | 0.4382 |
| FrameBridge (Ours, w/o SAF) | 433 | 38.61 | 0.5989 | 229 | 0.2246 | 0.5559 |
| FrameBridge (Ours, w/ SAF) | 312 | **39.89**$^‡$ | 0.6697 | **99**$^‡$ | **0.2250**$^†$ | 0.6963 |
| FrameBridge-100k (Ours, w/SAF) | 258 | **44.13**$^†$ | **0.7274**$^‡$ | **95**$^†$ | **0.2250**$^†$ | **0.7142**$^‡$ |

Table 2: VBench-I2V (Huang et al., 2024) scores for different I2V models. For all the evaluation dimensions, higher score means better performance. For results marked by $*$, we directly use the data of VBench-I2V Leaderboard. For each metric, we mark the best one with † and the second one with ‡. The abbreviations represents Subject Consistency (SC), Background Consistency (BC), Temporal Flickering (TF), Motion Smoothness (MS), Dynamic Degree (DD), Aesthetic Quality (AQ), Imaging Quality (IQ).

| Model | SC | BC | TF | MS | DD | AQ | IQ |
|---|---|---|---|---|---|---|---|
| DynamiCrafter | **95.40**$^‡$ | 96.22 | 97.03 | **97.82**$^‡$ | **38.69**$^‡$ | **59.40**$^†$ | 62.29 |
| SEINE | 93.45 | 94.21 | 95.07 | 96.20 | 24.55 | 56.55 | **70.52**$^†$ |
| SparseCtrl | 88.39 | 92.46 | 91.78 | 94.25 | **81.95**$^†$ | 49.88 | **69.35**$^‡$ |
| ConsistI2V$^*$ | 95.27 | **98.28**$^†$ | **97.56** | 97.38 | 18.62 | 59.00 | 66.92 |
| FrameBridge | **96.24**$^†$ | **97.25**$^‡$ | **98.01**$^†$ | **98.51**$^†$ | 35.77 | **59.38**$^‡$ | 63.28 |

## 4.3 SNR-ALIGNED FINE-TUNING

To implement FrameBridge, one straightforward strategy is to train bridge models from scratch using eq. (6). Meanwhile, another common practice of training I2V models is to fine-tune from pre-trained T2V diffusion models (Chen et al., 2023b;a; Xing et al., 2023; Blattmann et al., 2023; Ma et al., 2024a). A key challenge arises as the marginal distribution of bridge models $p_{t,bridge}(\mathbf{z}_t)$ differs from that of diffusion models $p_{t,diff}(\mathbf{z}_t)$, limiting the generation performance of fine-tuned FrameBridge as illustrated in Figure 3. To address this issue, we propose the innovative SNR-Aligned Fine-tuning (SAF) technique. By bridging the gap between the two distributions during fine-tuning, SAF enables more seamless knowledge transfer between the two model families. To the best of our knowledge, this is the first attempt to explore how to effectively fine-tune bridge models from pre-trained diffusion models. Our SAF technique consists of the following two steps:

**Reparameterization of Bridge Process.** In bridge process, the perturbed latent $\mathbf{z}_t$ at timestep $t$ can be written as the linear combination of $\mathbf{z}_0$, $\mathbf{z}_T$ and a Gaussian noise $\boldsymbol{\epsilon}$: $\mathbf{z}_t = a_t \mathbf{z}_0 + b_t \mathbf{z}_T + c_t \boldsymbol{\epsilon}$ (detailed expression of $a_t, b_t, c_t$ can be found in eq. (12)), which is different from $\alpha_t \mathbf{z}_0 + \sigma_t \boldsymbol{\epsilon}$[1] in diffusion models due to the change of prior and forward process. Therefore, the pre-trained diffusion models have limited ability to directly denoise such a $\mathbf{z}_t$, which impairs effective fine-tuning. To match the distributions of $\mathbf{z}_t$, we reparameterize the bridge process by

$$\tilde{\mathbf{z}}_t = \frac{\mathbf{z}_t - b_t \mathbf{z}_T}{\sqrt{a_t^2 + c_t^2}} = \frac{a_t}{\sqrt{a_t^2 + c_t^2}} \mathbf{z}_0 + \frac{c_t}{\sqrt{a_t^2 + c_t^2}} \boldsymbol{\epsilon}. \tag{7}$$

---

[1] Without loss of generality, we consider Variance-Preserving (VP) diffusion where $\alpha_t^2 + \sigma_t^2 = 1$

Then, $\tilde{\mathbf{z}}_t$ can be represented as the combination of clean data $\mathbf{z}_0$ and a Gaussian noise, with the squre sum of coefficients equal to 1. Thus, the reparameterized bridge process $\tilde{\mathbf{z}}_t$ exactly align with a VP diffusion process,

**SNR-based Latent Alignment**    Although $\tilde{\mathbf{z}}_t$ has the same perturbation kernel as a diffusion process, the marginal distribution of $\tilde{\mathbf{z}}_t$ does not match that of the pre-trained diffusion at timestep $t$, *i.e.*, $\alpha_t \mathbf{z}_0 + \sigma_t \epsilon$, hindering bridge models to seamlessly leverage the knowledge of pre-trained diffusion models (see Figure 3). Therefore, we leverage signal-to-noise ratio (SNR) as an indicator to unify the two marginal distributions (Kingma et al., 2021). Specifically, we change the timestep $t$ to another $\tilde{t}$ such that $\alpha_{\tilde{t}} = \frac{a_t}{\sqrt{a_t^2 + c_t^2}}$, $\sigma_{\tilde{t}} = \frac{c_t}{\sqrt{a_t^2 + c_t^2}}$, and then $\tilde{\mathbf{z}}_t$ has the same SNR as $\alpha_{\tilde{t}} \mathbf{z}_0 + \sigma_{\tilde{t}} \epsilon$ in diffusion process. According to the above derivation, we reparameterize the input of bridge models as $\epsilon_{\theta,bridge}^{\hat{\Psi}}(\mathbf{z}_t, t, i, c) \triangleq \epsilon_{\theta,aligned}^{\hat{\Psi}}(\tilde{\mathbf{z}}_t, \tilde{t}, i, c)$, and initialize $\epsilon_{\theta,aligned}^{\hat{\Psi}}$ with the pre-trained T2V diffusion models. Since the marginal distribution of $\tilde{\mathbf{z}}_t$ is complemetely aligned with the marginal of diffusion process at timestep $\tilde{t}$, SAF enables bridge models to fully exploit the denoising capability of pre-trained diffusion models. We provide a general statement of SAF technique and more details in Appendix A.

## 4.4 NEURAL PRIOR

By establishing a *data-to-data* process for I2V synthesis, we have been able to reduce the distance between the prior and the target from *noise-to-frames* to *frames-to-frames*, and therefore reduce the burden of generative models and aim at improving synthesis quality. To further demonstrate the function of improving prior information for I2V synthesis, we extend our design of FrameBridge from replicated initial frame $\mathbf{z}^i$ to neural representations $F_\eta(z^i, c)$, which serves as a stronger prior for video frames.

As shown in Figure 4, although the static frame has provided indicative information such as the appearance details of the background and different objects, it may not be informative for the motion information in consecutive frames. When the distance between the prior frame and the target frame is large, bridge models are faced with the challenge to generate the motion trajectory. Therefore, we present a stronger prior than simply duplicating the initial frame, *neural prior*, which achieves a coarse estimation of the target at first, and then bridge models generate the high-quality target from this coarse estimation.

Considering bridge models synthesize target data with iterative sampling steps, we develop a one-step mapping-based prior network taking both image latent $z^i$ and text or label condition $c$ as input, and separately train the prior network with a regression loss in latent space:

$$\mathcal{L}_{prior}(\eta) = \mathbb{E}_{(\mathbf{z}, z^i, c) \sim p_{data}(\mathbf{z}, z^i, c)} \left[ \left\| F_\eta(z^i, c) - \mathbf{z} \right\|^2 \right]. \tag{8}$$

With this objective, it can be proved that $F_\eta(z^i, c)$ learns to predict the mean value of subsequent frames, as shown in Appendix A). Given pre-trained $F_\eta(z^i, c)$, we build FrameBridge-NP from its output and target video latent $\mathbf{z}$ by replacing the prior $\mathbf{z}_T$ in eq. (6) with the neural prior $F_\eta(z^i, c)$. More details of the training and sampling algorithm can be found in Appendix E.

In generation, neural prior model $F_\eta(z^i, c)$ provide a coarse estimation with a single deterministic step, which is closer to the target than the provided initial frame, and bridge model synthesize the video target with a coarse-to-fine iterative sampling process. Although more advanced methods can be designed to further improve neural prior, we present a design with simple training objective and one-step sampling, demonstrating the performance of enhancing prior information on I2V synthesis.

Table 3: Non-zero-shot I2V generation on UCF-101. Different from zero-shot metrics, here all models are trained on UCF-101 dataset. The best and second results are marked with † and ‡. The PIC value of ExtDM is not comparable to other methods as its resolution $64 \times 64$ is much lower than others and it generate videos with slower motion.

| Method | FVD ↓ | IS ↑ | PIC ↑ |
|---|---|---|---|
| ExtDM | 649 | 21.37 | **0.9651**[†] |
| VDT-I2V | 171 | 62.61 | 0.7401 |
| FrameBridge | **154**[‡] | **64.01**[†] | 0.7443 |
| FrameBridge-NP | **122**[†] | 63.60[‡] | 0.7662[‡] |

# 5    EXPERIMENTS

We carry out experiments on UCF-101 (Soomro, 2012) and WebVid-2M (Bain et al., 2021) datasets to demonstrate the advantages of our data-to-data generation framework for I2V tasks.

## 5.1    SETUPS

**Class-conditional I2V Generation**    For class-conditional I2V generation, we train FrameBridge and other baselines on UCF-101 dataset. Fréchet Video Distance (Unterthiner et al. (2018); FVD) and Inception Score (Saito et al. (2017); IS) are used to evaluate the quality of generated videos. Meanwhile, we use Perceptual Input Conformity (PIC) (Xing et al., 2023) to evaluate the consistency of the synthesized frames with given image conditions.

**Text-conditional I2V Generation**    For text-conditional I2V generation, FrameBridge models are fine-tuned from pre-trained T2V diffusion models on WebVid-2M dataset. We use zero-shot FVD, IS, PIC on UCF-101 and zero-shot FVD, CLIPSIM (Wu et al., 2021), PIC on MSR-VTT (Xu et al., 2016) to evalaute the quality and consistency with image conditions for generated videos. More details about datasets and metrics can be found in Appendix C.

## 5.2    FINE-TUNING FROM PRE-TRAINED DIFFUSION MODELS

**Implementation Details**    Following (Xing et al., 2023), we fine-tune FrameBridge with the replicated prior $\mathbf{z}^i$ from the open-sourced T2V diffusion model VideoCrafter1 (Chen et al., 2023a). When applying SAF technique proposed in Section 4.3, we initialize the network after alignment $\boldsymbol{\epsilon}_{\theta,aligned}^{\hat{\Psi}}(\mathbf{z}_t, t, z^i, c)$ with the weights of VideoCrafter1 at $256 \times 256$ resolutions, otherwise we directly initialize the bridge model $\boldsymbol{\epsilon}_\theta^{\hat{\Psi}}(\mathbf{z}_t, t, z^i, c)$ with VideoCrafter1. We fine-tune two FrameBridge models (with and without SAF) for 20k iterations with batch size 64, and fine-tune a FrameBridge model with SAF for 100k iterations (FrameBridge-100k). All synthesized videos are sampled through the first-order SDE solvers with 50 steps (Chen et al., 2023c). More details can be found in Appendix C.

**Comparison with Baselines**    We choose Dynami-Crafter (Xing et al., 2023), SEINE (Chen et al., 2023b), I2VGen-XL (Zhang et al., 2023), SVD (Blattmann et al., 2023), ConsistI2V (Ren et al., 2024) and SparseCtrl (Guo et al., 2025) as text-conditional I2V baselines. Table 1 shows zero-shot metrics on UCF-101 and MSR-VTT after fine-tuning on WebVid-2M. We also evaluate FrameBridge and other baselines with a comprehensive benchmark for video quality, *i.e.*, VBench-I2V (Huang et al., 2024) [2]. As demonstrated by quantitative results in the tables, FrameBridge can effectively leverage the

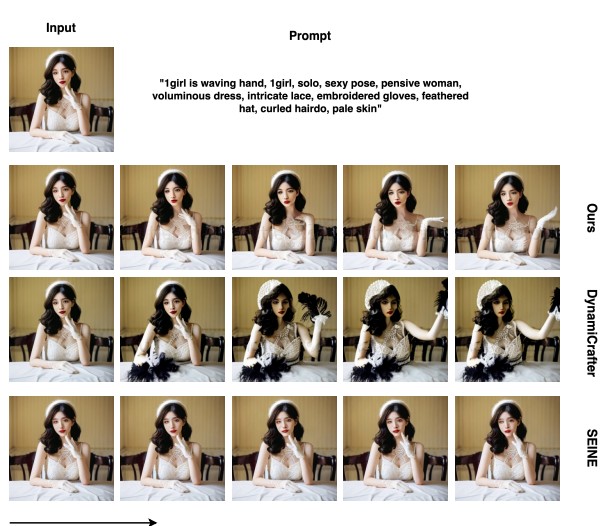

Figure 5: **Qualitative comparisons between Frame-Bridge and other baselines.** FrameBridge outperforms other diffusion-based methods in appearance consistency and video quality.

knowledge from pre-trained T2V diffusion models and generate videos with higher quality and consistency than the diffusion counterparts. Moreover, it is indicated that SAF can further improve the performance of fine-tuned Frame-Bridge, showcasing the advantages of matching marginal distributions of the bridge process and diffusion process. We further discuss the trade-off between the dynamic degree and consistency in Appendix D.1. Qualitative results are shown in Figure 5.

---

[2]For some models, we directly use the evaluation results on VBench Leaderboard `https://huggingface.co/spaces/Vchitect/VBench_Leaderboard/`.

Table 4: Ablation of SAF technique on UCF-101. All models are fine-tuned from Latte-XL/2 class-conditional video diffusion model for 20k iterations. We use repeating regression for FrameBridge, and w/o SAF means fine-tuning without SAF technique.

| Method | FVD ↓ | IS ↑ | PIC ↑ |
|---|---|---|---|
| VDT-I2V | 177 | 55.22 | 0.7700 |
| FrameBridge(w/o SAF) | 157 | 37.65 | 0.7498 |
| FrameBridge(w/ SAF) | 148 | 55.60 | 0.8198 |

Table 5: Ablation of neural prior. Condition means whether the model condition on $F_\eta(z^i, c)$. We use FVD to evaluate the video quality.

| Method | Prior | Condition | FVD ↓ |
|---|---|---|---|
| VDT-I2V | Gaussian | ✗ | 171 |
| VDT-I2V | Gaussian | ✓ | 132 |
| FrameBridge | replicated | ✗ | 154 |
| FrameBridge | replicated | ✓ | 129 |
| FrameBridge-NP | neural | ✓ | 122 |

Both quantitative and qualitative comparisons demonstrate the advantages of FrameBridge and the data-to-data generation framework for I2V tasks. To the best of our knowledge, our trial is the first time to fine-tune diffusion bridge models from pre-trained diffusion models, and SAF is crucial to boosting the performance of fine-tuned bridge models.

### 5.3 NEURAL PRIOR FOR BRIDGE MODELS

**Implementation Details**  We implement FrameBridge-NP based on the class-conditional video diffusion model Latte-S/2 (Ma et al., 2024b) by replacing diffusion process with the Bridge-gmax bridge process. We freeze the parameters $\eta$ of neural prior when training bridge models $\epsilon_\theta^{\hat{\Psi}}(\mathbf{z}_t, t, \mathbf{z}_T, z^i, c)$. All synthesized videos are sampled through the first-order SDE solvers with 250 steps. More details can be found in Appendix C.

**Comparison with Baselines**  We reproduce two diffusion models ExtDM (Zhang et al., 2024b) and VDT (Lu et al., 2023) on UCF-101 dataset for the class-conditional I2V task as our baselines. Table 3 shows that FrameBridge-NP has superior video quality and consistency with condition images. More qualitative results are shown in Appendix F. The experiments reveal that bridge-based I2V models outperform their diffusion counterparts with both replicated prior and neural prior, justifying the usage of the *data-to-data* generation process for I2V tasks. Additionally, FrameBridge can further benefit from neural prior $F_\eta(z^i, c)$ as it actually narrows the gap between the prior and data distribution of bridge process.

### 5.4 ABLATION STUDIES

**SNR-Aligned Fine-tuning**  Table 1 has already shown the advantage of SAF technique, we use another ablation experiment to further elucidate that. We fine-tune a pre-trained class-conditional video generation model Latte-XL/2[3] on UCF-101 with 20k iterations. The quantitative results presented in Table 4 shows that SAF improves generation performance of FrameBridge.

**Neural Prior**  To showcase the effectiveness of neural prior, we compare five different models varying in priors and network conditions. More details of the configurations can be found in Appendix C. Results in Table 5 reveal that $F_\eta(z^i, c)$ is indeed more informative than a single frame $z^i$ and can be fully utilized by FrameBridge through the change of prior.

## 6 CONCLUSIONS

In this work, we propose FrameBridge, building a *data-to-data* generation process for I2V synthesis, which matches the *frame-to-frames* nature of this task. Additionally, targeting at two typical scenarios of training I2V models, namely fine-tuning from pre-trained diffusion models and training from scratch, we present SNR-Aligned Fine-tuning (SAF) and neural prior respectively to further improve the generation quality of FrameBridge. Extensive experiments show that FrameBridge generate videos with enhanced appearance consistency with image condition and improved temporal coherence between frames, demonstrating the advantages of FrameBridge over diffusion-based I2V methods and the effectiveness of two proposed techniques.

---

[3]https://huggingface.co/maxin-cn/Latte/tree/main

# 7 REPRODUCIBILITY STATEMENT

To ensure the reproducibility of FrameBridge, we elaborate the reference works, codebase, datasets, pre-trained model checkpoints. The code for fine-tuning and training FrameBridge models are based on the open-sourced codebase of DDBM[4], Latte[5] and DynamiCrafter[6]. Details about UCF-101, WebVid-2M, MSR-VTT datasets can be found in C. When fine-tuning I2V models from pre-trained diffusion models, we use pre-trained checkpoints of VideoCrafter1 to initialize the weights of our model, which is available at `https://huggingface.co/VideoCrafter/Text2Video-256`. In Appendix C, we present details for reproducing the training and evaluation process of Frame-Bridge. The proof of claims are provided in A. Additionally, we provide enough demo samples on our demo page and our code will also be released there upon acceptance.

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

# A  PROOF AND DERIVATION

## A.1  BASICS OF DENOISING DIFFUSION BRIDGE MODEL (DDBM)

We provide the derivations of $p_{t,bridge}(\mathbf{z}_t|\mathbf{z}_0, \mathbf{z}_T, z^i, c)$ and $h(\mathbf{z}, t, \mathbf{y}, z^i, c)$ used in Section 4.2.

Similar to the proofs in (Zhou et al., 2023), we calculate $p_{t,bridge}(\mathbf{z}_t|\mathbf{z}_0, \mathbf{z}_T, z^i, c)$ by applying Bayes' rule:

$$
\begin{aligned}
p_{t,bridge}(\mathbf{z}_t|\mathbf{z}_0, \mathbf{z}_T, z^i, c) = p_{t,diff}(\mathbf{z}_t|\mathbf{z}_0, \mathbf{z}_T, z^i, c) &= \frac{p_{T,diff}(\mathbf{z}_T|\mathbf{z}_t, \mathbf{z}_0, z^i, c)p_{t,diff}(\mathbf{z}_t|\mathbf{z}_0, z^i, c)}{p_{t,diff}(\mathbf{z}_T|\mathbf{z}_0, z^i, c)} \\
&\overset{①}{=} \frac{p_{T,diff}(\mathbf{z}_T|\mathbf{z}_t)p_{t,diff}(\mathbf{z}_t|\mathbf{z}_0)}{p_{T,diff}(\mathbf{z}_T|\mathbf{z}_0)}.
\end{aligned}
\tag{9}
$$

① uses the Markovian of the diffusion process $\mathbf{z}_t$ (Kingma et al., 2021).

The perturbation kernels $p_{T,diff}(\mathbf{z}_T|\mathbf{z}_t), p_{t,diff}(\mathbf{z}_t|\mathbf{z}_0), p_{T,diff}(\mathbf{z}_T|\mathbf{z}_0)$ is Gaussian and takes the form of:

$$
\begin{aligned}
p_{T,diff}(\mathbf{z}_T|\mathbf{z}_t) &= \mathcal{N}(\mathbf{z}_T; \frac{\alpha_T}{\alpha_t}\mathbf{z}_t, (\sigma_T^2 - \frac{\alpha_T^2}{\alpha_t^2}\sigma_t^2)I), \\
p_{t,diff}(\mathbf{z}_t|\mathbf{z}_0) &= \mathcal{N}(\mathbf{z}_t; \alpha_t\mathbf{z}_0, \sigma_t^2 I), \\
p_{T,diff}(\mathbf{z}_T|\mathbf{z}_0) &= \mathcal{N}(\mathbf{z}_T; \alpha_T\mathbf{z}_0, \sigma_T^2 I).
\end{aligned}
\tag{10}
$$

Following (Zhou et al., 2023), it can be derived that $p_{t,bridge}(\mathbf{z}_t|\mathbf{z}_T, \mathbf{z}_0, z^i, c)$ is also Gaussian, and $p_{t,bridge}(\mathbf{z}_t|\mathbf{z}_T, \mathbf{z}_0, z^i, c) = \mathcal{N}(\mathbf{z}_t; \mu_t(\mathbf{z}_0, \mathbf{z}_T), \sigma_{t,bridge}^2 I)$, where

$$
\begin{aligned}
\mu_t(\mathbf{z}_0, \mathbf{z}_T) &= \alpha_t(1 - \frac{\text{SNR}_T}{\text{SNR}_t})\mathbf{z}_0 + \frac{\text{SNR}_T}{\text{SNR}_t}\frac{\alpha_t}{\alpha_T}\mathbf{z}_T, \\
\sigma_{t,bridge}^2 &= \sigma_t^2(1 - \frac{\text{SNR}_T}{\text{SNR}_t}).
\end{aligned}
\tag{11}
$$

Specifically, $\mathbf{z}_t$ of bridge process can be reparameterized by $\mathbf{z}_t = a_t\mathbf{z}_0 + b_t\mathbf{z}_T + c_t\boldsymbol{\epsilon}$, where

$$
\begin{aligned}
a_t &= \alpha_t(1 - \frac{\text{SNR}_T}{\text{SNR}_t}), \\
b_t &= \frac{\text{SNR}_T}{\text{SNR}_t}\frac{\alpha_t}{\alpha_T}, \\
c_t &= \sqrt{\sigma_t^2(1 - \frac{\text{SNR}_T}{\text{SNR}_t})}.
\end{aligned}
\tag{12}
$$

Here, $\text{SNR}_t = \frac{\alpha_t^2}{\sigma_t^2}$ (Kingma et al., 2021) is the signal-to-noise ratio of diffusion process.

Then we calculate $h(\mathbf{z}, t, \mathbf{y}, z^i, c) = \nabla_{\mathbf{z}_t} \log p_{T,diff}(\mathbf{z}_T|\mathbf{z}_t)|_{\mathbf{z}_t=\mathbf{z}, \mathbf{z}_T=\mathbf{y}}$.

As $p_{T,diff}(\mathbf{z}_T|\mathbf{z}_t) = \mathcal{N}(\mathbf{z}_T; \frac{\alpha_T}{\alpha_t}\mathbf{z}_t, (\sigma_T^2 - \frac{\alpha_T^2}{\alpha_t^2}\sigma_t^2)I)$, we have

$$
p_{T,diff}(\mathbf{z}_T|\mathbf{z}_t) = \frac{1}{\sqrt{2\pi(\sigma_T^2 - \frac{\alpha_T^2}{\alpha_t^2}\sigma_t^2)}^D} \exp\left( -\frac{\left\| \mathbf{z}_T - \frac{\alpha_T}{\alpha_t}\mathbf{z}_t \right\|^2}{2(\sigma_T^2 - \frac{\alpha_T^2}{\alpha_t^2}\sigma_t^2)} \right),
\tag{13}
$$

$$
\log p_{T,diff}(\mathbf{z}_T|\mathbf{z}_t) = -\frac{\left\| \mathbf{z}_T - \frac{\alpha_T}{\alpha_t}\mathbf{z}_t \right\|^2}{2(\sigma_T^2 - \frac{\alpha_T^2}{\alpha_t^2}\sigma_t^2)} + C,
\tag{14}
$$

where $C$ is a constant independent of $\mathbf{z}_T$.

$$\nabla_{\mathbf{z}_t} \log p_{T,diff}(\mathbf{z}_T|\mathbf{z}_t) = \nabla_{\mathbf{z}_t} \left( -\frac{\left\| \mathbf{z}_T - \frac{\alpha_T}{\alpha_t}\mathbf{z}_t \right\|^2}{2(\sigma_T^2 - \frac{\alpha_T^2}{\alpha_t^2}\sigma_t^2)} \right) = -\frac{\mathbf{z}_T - \frac{\alpha_T}{\alpha_t}\mathbf{z}_t}{(\sigma_T^2 - \frac{\alpha_T^2}{\alpha_t^2}\sigma_t^2)}. \tag{15}$$

So, $\boldsymbol{h}(\mathbf{z}, t, \mathbf{y}, z^i, c) = -\frac{\mathbf{y} - \frac{\alpha_T}{\alpha_t}\mathbf{z}}{(\sigma_T^2 - \frac{\alpha_T^2}{\alpha_t^2}\sigma_t^2)}$. Note that for the diffusion process we commonly use, $\frac{\alpha_T}{\alpha_t} \approx 0$ and $\sigma_T \approx 1$, and we have $\boldsymbol{h}(\mathbf{z}, t, \mathbf{y}, z^i, c) \approx -\mathbf{y}$.

### A.2 PARAMETERIZATION OF FRAMEBRIDGE

**Proposition 1.** *The score estimation $\boldsymbol{s}_\theta(\mathbf{z}_t, t, \mathbf{z}_T, z^i, c)$ of bridge process $p_{t,bridge}(\mathbf{z}_t|\mathbf{z}_T, z^i, c)$ can be reparamterized by*

$$\boldsymbol{s}_\theta(\mathbf{z}_t, t, \mathbf{z}_T, z^i, c) = -\frac{1}{\sigma_t}\boldsymbol{\epsilon}_\theta^{\hat{\Psi}}(\mathbf{z}_t, t, \mathbf{z}_T, z^i, c) - \frac{\mathrm{SNR}_T}{\mathrm{SNR}_t}\frac{\mathbf{z}_t - \frac{\alpha_t}{\alpha_T}\mathbf{z}_T}{\sigma_t^2(1 - \frac{\mathrm{SNR}_T}{\mathrm{SNR}_t})}, \tag{16}$$

*where $\mathrm{SNR}_t = \frac{\alpha_t^2}{\sigma_t^2}$, and $\boldsymbol{\epsilon}_\theta^{\hat{\Psi}}(\mathbf{z}_t, t, \mathbf{z}_T, z^i, c)$ is trained with the objective*

$$\mathcal{L}_{bridge}(\theta) = \mathbb{E}_{\substack{(\mathbf{z}_0, z^i, c) \sim p_{data}(\mathbf{z}_0, z^i, c), \\ \mathbf{z}_T = z^i, t, \mathbf{z}_t \sim p_{t,bridge}(\mathbf{z}_t|\mathbf{z}_0, \mathbf{z}_T, z^i, c)}} \left[ \tilde{\lambda}(t) \left\| \boldsymbol{\epsilon}_\theta^{\hat{\Psi}}(\mathbf{z}_t, t, \mathbf{z}_T, z^i, c) - \frac{\mathbf{z}_t - \alpha_t\mathbf{z}_0}{\sigma_t} \right\|^2 \right]. \tag{17}$$

*Here $\tilde{\lambda}(t)$ is the weight function of timestep $t$ and we take $\tilde{\lambda}(t) = 1$ unless otherwise specified.*

*When $\mathrm{SNR}_T \approx 0$(which is often the case for diffusion process), there exists $\epsilon$ such that*

$$\boldsymbol{s}_\theta(\mathbf{z}_t, t, \mathbf{z}_T, z^i, c) \approx -\frac{1}{\sigma_t}\boldsymbol{\epsilon}_\theta^{\hat{\Psi}}(\mathbf{z}_t, t, \mathbf{z}_T, z^i, c), \quad \forall t \in [\epsilon, T - \epsilon]. \tag{18}$$

*Proof.* We denote the desnoising target $\frac{\mathbf{z}_t - \alpha_t\mathbf{z}_0}{\sigma_t}$ by $\boldsymbol{\epsilon}^{\hat{\Psi}}(\mathbf{z}_t, \mathbf{z}_0, t)$, and define $a_t = \alpha_t(1 - \frac{\mathrm{SNR}_T}{\mathrm{SNR}_t})$, $b_t = \frac{\mathrm{SNR}_T}{\mathrm{SNR}_t}\frac{\alpha_t}{\alpha_T}$, $c_t = \sqrt{\sigma_t^2(1 - \frac{\mathrm{SNR}_T}{\mathrm{SNR}_t})}$.

From eq. (11), we have

$$\nabla_{\mathbf{z}} \log p_{t,bridge}(\mathbf{z}|\mathbf{z}_0, \mathbf{z}_T)|_{\mathbf{z}=\mathbf{z}_t, \mathbf{z}_T=\mathbf{z}^i} = -\frac{\mathbf{z}_t - a_t\mathbf{z}_0 - b_t\mathbf{z}^i}{c_t^2}, \tag{19}$$

which is the target of Denoising Bridge Score Matching (Zhou et al., 2023). Our goal is to represent this target with $\mathbf{z}_t$, $\mathbf{z}_T$, and $\boldsymbol{\epsilon}^{\hat{\Psi}}(\mathbf{z}_t, \mathbf{z}_0, t)$.

From the definition of $\boldsymbol{\epsilon}^{\hat{\Psi}}(\mathbf{z}_t, \mathbf{z}_0, t)$, we have

$$\mathbf{z}_0 = \frac{\mathbf{z}_t - \sigma_t\boldsymbol{\epsilon}^{\hat{\Psi}}(\mathbf{z}_t, \mathbf{z}_0, t)}{\alpha_t}. \tag{20}$$

Plug it into eq. (19), it can be derived that

$$\nabla_{\mathbf{z}} \log p_{t,bridge}(\mathbf{z}|\mathbf{z}_0, \mathbf{z}_T)|_{\mathbf{z}=\mathbf{z}_t, \mathbf{z}_T=\mathbf{z}^i} = -\frac{\mathbf{z}_t - a_t \frac{\mathbf{z}_t - \sigma_t \boldsymbol{\epsilon}^{\hat{\Psi}}(\mathbf{z}_t, \mathbf{z}_0, t)}{\alpha_t} - b_t \mathbf{z}^i}{c_t^2}$$

$$= -\frac{\alpha_t \mathbf{z}_t - a_t \mathbf{z}_t + a_t \sigma_t \boldsymbol{\epsilon}^{\hat{\Psi}}(\mathbf{z}_t, \mathbf{z}_0, t) - \alpha_t b_t \mathbf{z}_T}{\alpha_t c_t^2}$$

$$= -\frac{a_t \sigma_t \boldsymbol{\epsilon}^{\hat{\Psi}}(\mathbf{z}_t, \mathbf{z}_0, t)}{\alpha_t c_t^2} - \frac{(\alpha_t - a_t)\mathbf{z}_t - \alpha_t b_t \mathbf{z}^i}{\alpha_t c_t^2} \qquad (21)$$

$$= -\frac{1}{\sigma_t}\boldsymbol{\epsilon}^{\hat{\Psi}}(\mathbf{z}_t, \mathbf{z}_0, t) - \frac{\alpha_t \frac{\mathrm{SNR}_T}{\mathrm{SNR}_t}\mathbf{z}_t - \frac{\alpha_t^2}{\alpha_T}\frac{\mathrm{SNR}_T}{\mathrm{SNR}_t}\mathbf{z}^i}{\alpha_t \sigma_t^2 (1 - \frac{\mathrm{SNR}_T}{\mathrm{SNR}_t})}$$

$$= -\frac{1}{\sigma_t}\boldsymbol{\epsilon}^{\hat{\Psi}}(\mathbf{z}_t, \mathbf{z}_0, t) - \frac{\mathrm{SNR}_T}{\mathrm{SNR}_t}\frac{\mathbf{z}_t - \frac{\alpha_t}{\alpha_T}\mathbf{z}^i}{\sigma_t^2 (1 - \frac{\mathrm{SNR}_T}{\mathrm{SNR}_t})},$$

As the Denoising Bridge Score Matching takes the form of

$$\mathcal{L}_{bridge}(\theta) = \mathbb{E}_{(\mathbf{z}_0, z^i, c).\mathbf{z}_T = \mathbf{z}^i, t, \mathbf{z}_t}\left[\lambda(t)\left\|\boldsymbol{s}_\theta(\mathbf{z}_t, t, \mathbf{z}_T, z^i, c) - \nabla_{\mathbf{z}}\log p_{t,bridge}(\mathbf{z}|\mathbf{z}_0, \mathbf{z}_T)|_{\mathbf{z}=\mathbf{z}_t, \mathbf{z}_T=\mathbf{z}^i}\right\|^2\right], \qquad (22)$$

when we parameterize $\boldsymbol{s}_\theta(\mathbf{z}_t, t, \mathbf{z}_T, z^i, c) = -\frac{1}{\sigma_t}\boldsymbol{\epsilon}_\theta^{\hat{\Psi}}(\mathbf{z}_t, t, \mathbf{z}_T, z^i, c) - \frac{\mathrm{SNR}_T}{\mathrm{SNR}_t}\frac{\mathbf{z}_t - \frac{\alpha_t}{\alpha_T}\mathbf{z}_T}{\sigma_t^2(1 - \frac{\mathrm{SNR}_T}{\mathrm{SNR}_t})}$, the training objective can be written as

$$\mathcal{L}_{bridge}(\theta) = \mathbb{E}_{(\mathbf{z}_0, z^i, c).\mathbf{z}_T = \mathbf{z}^i, t, \mathbf{z}_t}\left[\frac{\lambda(t)}{\sigma_t^2}\left\|\boldsymbol{\epsilon}_\theta^{\hat{\Psi}}(\mathbf{z}_t, t, \mathbf{z}_T, z^i, c) - \boldsymbol{\epsilon}^{\hat{\Psi}}(\mathbf{z}_t, \mathbf{z}_0, t)\right\|^2\right], \qquad (23)$$

which proves the first part of the proposition if we take $\tilde{\lambda}(t) = \frac{\lambda(t)}{\sigma_t^2}$.

For the second part, when $\mathrm{SNR}_T \approx 0$, there exists an $\epsilon > 0$, such that $\frac{1}{\sigma_t^2(1 - \frac{\mathrm{SNR}_T}{\mathrm{SNR}_t})}$ has an upper bound $M$. Since $\frac{\mathrm{SNR}_T}{\mathrm{SNR}_t}\frac{\alpha_t}{\alpha_T} = \alpha_T \frac{\sigma_t^2}{\alpha_t \sigma_T^2} \approx 0$ when $\mathrm{SNR}_T \approx 0$, it can be directly inferenced from eq. (16) that $\boldsymbol{s}_\theta(\mathbf{z}_t, t, \mathbf{z}_T, z^i, c) \approx -\frac{1}{\sigma_t}\boldsymbol{\epsilon}_\theta^{\hat{\Psi}}(\mathbf{z}_t, t, \mathbf{z}_T, z^i, c)$. $\qquad \square$

**Remark.** *From the first part of the proposition, we parameterize bridge models to predict $\frac{\mathbf{z}_t - \alpha_t \mathbf{z}_0}{\sigma_t}$. It is similar to that used in Chen et al. (2023c) although their parameterization is derived from the forward-backward diffusion process of Schrödinger Bridge problems. The statement and proof of this proposition reveals that DDBM and Diffusion Schrödinger Bridges are closely related. Additionally, the second part shows that our parameterization resembles the Denoising Score Matching in diffusion models.*

### A.3   SNR-ALIGNED FINE-TUNING

**Existence and Uniqueness of $\tilde{t}$**   In Section 4.3, we need to find a $\tilde{t}$ such that $\alpha_{\tilde{t}} = \frac{a_t}{\sqrt{a_t^2 + c_t^2}}, \sigma_{\tilde{t}} = \frac{c_t}{\sqrt{a_t^2 + c_t^2}}$. Since $\frac{a_t^2}{c_t^2} = \frac{\alpha_t^2}{\sigma_t^2}(1 - \frac{\mathrm{SNR}_T}{\mathrm{SNR}_t}) = \mathrm{SNR}_t - \mathrm{SNR}_T$, it is a monotonically decreasing function of $t$. As $\mathrm{SNR}_t$ is also a monotonically decreasing function which ranges over $(0, \infty)$, we can take $\tilde{t} = \mathrm{SNR}^{-1}(\frac{a_t^2}{c_t^2})$ and the uniqueness of such $\tilde{t}$ can also be guaranteed. Next, we provide a more general form of SAF, where the schedule $\{\alpha_t, \sigma_t\}_{t \in [0,T]}$ of the pre-trained diffusion models and bridge models are not necessarily the same.

**Proposition 2.** *Suppose we fine-tune a Gaussian diffusion model $\tilde{\boldsymbol{\epsilon}}_\eta(\mathbf{z}_t, t, c)$ with schedule $\{\tilde{\alpha}_t, \tilde{\sigma}_t\}_{t \in [0,T]}$ to a diffusion bridge model $\boldsymbol{\epsilon}_{\theta,bridge}^{\hat{\Psi}}(\mathbf{z}_t, t, \mathbf{z}_T, z^i, c) \triangleq \boldsymbol{\epsilon}_{\theta,align}^{\hat{\Psi}}(\tilde{\mathbf{z}}_t, \tilde{t}, \mathbf{z}_T, z^i, c)$ with schedule $\{\alpha_t, \sigma_t\}_{t \in [0,T]}$. If we use the same dataset $p_{data}(\mathbf{z}_0, z^i, c)$ for training $\tilde{\boldsymbol{\epsilon}}_\eta(\mathbf{z}_t, t, c)$ and fine-tuning $\boldsymbol{\epsilon}_{\theta,align}^{\hat{\Psi}}(\tilde{\mathbf{z}}_t, \tilde{t}, \mathbf{z}_T, z^i, c)$. Then, for each $c$, the*

*input* $(\mathbf{z}_t, t)$ *of* $\tilde{\epsilon}_\eta$ *has the same marginal distribution as the input* $(\tilde{\mathbf{z}}_t, \tilde{t})$ *of* $\epsilon_{\theta,align}^{\hat{\Psi}}(\tilde{\mathbf{z}}_t, \tilde{t}, \mathbf{z}_T, z^i, c)$. *Here*

$$\tilde{\mathbf{z}}_t = \frac{\mathbf{z}_t - b_t \mathbf{z}^i}{\sqrt{a_t^2 + c_t^2}},$$

$$\tilde{t} = \widetilde{SNR}^{-1}(\frac{a_t^2}{c_t^2}). \tag{24}$$

$(\widetilde{SNR} = \frac{\tilde{\alpha}_t^2}{\tilde{\sigma}_t^2}$ *is the signal-to-noise ratio of pre-trained diffusion models.)*

*Proof.* Since $\widetilde{SNR}$ is also a monotonically decreasing function ranging over $(0, \infty)$, the uniqueness and existence of $\tilde{t}$ can also be guaranteed by the above analysis.

For a fixed $c, t$, we denote the probability density function of $\tilde{\mathbf{z}}_t$ by $q(\tilde{\mathbf{z}}_t; t)$. Then

$$\begin{aligned}
q(\tilde{\mathbf{z}}_t; t) &= \int_{z^i} q(\tilde{\mathbf{z}}_t | z^i; t) p_{data}(z^i) \mathrm{d}z^i \\
&= \int_{z^i} \int_{\mathbf{z}_0} q(\tilde{\mathbf{z}}_t | \mathbf{z}_0, z^i; t) p_{data}(\mathbf{z}_0, z^i) \mathrm{d}\mathbf{z}_0 \mathrm{d}z^i \\
&= \int_{z^i} \int_{\mathbf{z}_0} \mathcal{N}(\tilde{\mathbf{z}}_t; \frac{a_t}{\sqrt{a_t^2 + c_t^2}} \mathbf{z}_0, \frac{c_t}{\sqrt{a_t^2 + c_t^2}} I) p_{data}(\mathbf{z}_0, z^i) \mathrm{d}\mathbf{z}_0 \mathrm{d}z^i \\
&= \int_{z^i} \int_{\mathbf{z}_0} \mathcal{N}(\tilde{\mathbf{z}}_t; \tilde{\alpha}_{\tilde{t}} \mathbf{z}_0, \tilde{\sigma}_{\tilde{t}}^2 I) p_{data}(\mathbf{z}_0, z^i) \mathrm{d}\mathbf{z}_0 \mathrm{d}z^i \\
&= \int_{\mathbf{z}_0} \mathcal{N}(\tilde{\mathbf{z}}_t; \tilde{\alpha}_{\tilde{t}} \mathbf{z}_0, \tilde{\sigma}_{\tilde{t}}^2 I) (\int_{z^i} p_{data}(\mathbf{z}_0, z^i) \mathrm{d}z^i) \mathrm{d}\mathbf{z}_0 \\
&= \int_{\mathbf{z}_0} \mathcal{N}(\tilde{\mathbf{z}}_t; \tilde{\alpha}_{\tilde{t}} \mathbf{z}_0, \tilde{\sigma}_{\tilde{t}}^2 I) p_{data}(\mathbf{z}_0) \mathrm{d}\mathbf{z}_0,
\end{aligned} \tag{25}$$

which equals to the marginal distribution of the pre-trained diffusion process $p_{t,diff}(\mathbf{z}_t)$.

$\square$

### A.4 NEURAL PRIOR WITH REGRESSION TRAINING OBJECTIVE.

**Proposition 3.** *If we train* $F_\eta(z^i, c)$ *with the regression training objective*

$$\mathcal{L}_{prior}(\eta) = \mathbb{E}_{(\mathbf{z}_0, z^i, c) \sim p_{data}(\mathbf{z}_0, z^i, c)} \left[ \left\| F_\eta(z^i, c) - \mathbf{z}_0 \right\|^2 \right], \tag{26}$$

*and the neural network is optimized sufficiently, then we have*

$$F_\eta(z^i, c) = F_\eta^*(z^i, c) \triangleq \mathbb{E}_{\mathbf{z}_0 \sim p_{data}(\mathbf{z}_0 | z^i, c)} [\mathbf{z}_0]. \tag{27}$$

*Proof.* For each $(z^i, c)$, $\mathcal{L}_{prior}(\eta)$ optimizes the following objective:

$$\begin{aligned}
l_\eta(z^i, c) &= \mathbb{E}_{\mathbf{z}_0 \sim p_{data}(\mathbf{z}_0 | z^i, c)} \left[ \left\| F_\eta(z^i, c) - \mathbf{z}_0 \right\|^2 \right] \\
&= \left\| F_\eta(z^i, c) \right\|^2 - \langle F_\eta(z^i, c), \mathbb{E}_{\mathbf{z}_0 \sim p_{data}(\mathbf{z}_0 | z^i, c)} [\mathbf{z}_0] \rangle + \left\| \mathbb{E}_{\mathbf{z}_0 \sim p_{data}(\mathbf{z}_0 | z^i, c)} [\mathbf{z}_0] \right\|^2 \\
&= \left\| F_\eta(z^i, c) \right\|^2 - \langle F_\eta(z^i, c), \mathbb{E}_{\mathbf{z}_0 \sim p_{data}(\mathbf{z}_0 | z^i, c)} [\mathbf{z}_0] \rangle + C.
\end{aligned} \tag{28}$$

where $C$ is a constant independent of $\eta$. When the network is optimized sufficiently, $l_\eta(z^i, c)$ takes the minimum for each $(z^i, c)$, so we have

$$F_\eta(z^i, c) = \arg\min_{\mathbf{x}} \left( \|\mathbf{x}\|^2 - \langle \mathbf{x}, \mathbb{E}_{\mathbf{z}_0 \sim p_{data}(\mathbf{z}_0 | z^i, c)} [\mathbf{z}_0] \rangle \right) \tag{29}$$

It can be solved that $F_\eta(z^i, c) = \mathbb{E}_{\mathbf{z}_0 \sim p_{data}(\mathbf{z}_0 | z^i, c)} [\mathbf{z}_0]$.

$\square$

# B    DETAILED DISCUSSION ON RELATED WORKS

**Video Diffusion Models**    Inspired by the success of text-to-image (T2I) diffusion models (Ramesh et al., 2022; Nichol et al., 2021), numerous studies have investigated diffusion-based text-to-video (T2V) models (Blattmann et al., 2023; Yang et al., 2024; Singer et al., 2022) by designing 3D spatial-temporal U-Net (Ho et al., 2022b;a) and Diffusion Transformers (DiT) (Peebles & Xie, 2023; Bao et al., 2023). To improve memory and computation efficiency, Latent Diffusion Models (LDM) (Rombach et al., 2022; Vahdat et al., 2021) are utilized where the diffusion process is applied in the compressed latent space of video samples (Bao et al., 2024; Brooks et al., 2024; He et al., 2022). Meanwhile, some other works designed cascaded diffusion models to generate motion representation (Yu et al., 2024) or videos with lower resolution (Ho et al., 2022a; Wang et al., 2023) first, which are utilized to synthesize the result videos in the subsequent stages.

**Diffusion-based I2V Generation**    The main difference between I2V and T2V is the incorporation of image conditions into the sampling process. Xing et al. (2023) utilizes the features of a CLIP image encoder and a lightweight transformer to inject image conditions into the backbone of a T2V model. Ma et al. (2024a) and Zhang et al. (2024c) propose to directly model the residual between the subsequent frames and the given initial frame with diffusion for I2V generation. Moreover, Ma et al. (2024a) also uses the DCTInit technique to enhance the consistency of video content with the given image. Chen et al. (2023b) presents to train short-to-long video generation models with masked diffusion models. Guo et al. (2023) and Zhang et al. (2024a) propose to utilize pre-trained T2I models for image animation by training an additional component to model the relationship between video frames. SparseCtrl (Guo et al., 2025) and Animate Anyone (Hu, 2024) design specific fusion modules for video diffusion models to adapt to various types of conditions including RGB images. Ren et al. (2024) propose improved network architecture and sampling strategy for image-to-video generation at the same time to enhance the controllability of image conditions. Jain et al. (2024), Zhang et al. (2023) and Shi et al. (2024) design cascaded diffusion systems for I2V generation. VIDIM (Jain et al., 2024) consists of one base diffusion model and another two diffusion models for spatial and temporal super-resolution respectively. Zhang et al. (2023) uses a base diffusion model to generate videos with low resolutions, which serve as the input of the following video super-resolution diffusion model. Shi et al. (2024) first generates the optical flow between the subsequent frames and given image with a diffusion process, and use the optical flow as conditions of another model to generate videos. Ni et al. (2023) and Zhang et al. (2024b) train an autoencoder to represent the motions between frames in a latent space, and use diffusion models to generate motion latents. However, previous I2V diffusion models are built on the *noise-to-data* generation of conditional diffusion process and the sampling remains a denoising process conditioned on given images. In contrast, FrameBridge replaces the diffusion process with a bridge process and the sampling directly model the animation of static images.

**Noise Manipulation for Video Diffusion Models**    Several works have explored to improve the uninformative prior distribution of diffusion models. PYoCo (Ge et al., 2023) recently proposes to use correlated noise for each frame in both training and inference. ConsistI2V (Ren et al., 2024), FreeInit (Wu et al., 2023), and CIL (Zhao et al., 2024) present training-free strategies to better align the training and inference distribution of diffusion prior, which is popular in diffusion models (Lin et al., 2024; Podell et al., 2023; Blattmann et al., 2023; Girdhar et al., 2023). These strategies focus on improving the noise distribution to enhance the quality of synthesized videos, while they still suffer the restriction of noise-to-data diffusion framework, which may limit their endeavor to utilize the entire information (*e.g.*, both large-scale features and fine-grained details) contained in the given image. In contrast, we propose a *data-to-data* framework and utilize deterministic prior rather than Gaussian noise, allowing us to leverage the clean input image as prior information.

**Comparison with Previous Works of Bridge Models**    In Section 3, we leverage the forward SDE of bridge models (Zhou et al., 2023) and the backward sampler proposed by Chen et al. (2023c) to build FrameBridge. We unify their theoretical frameworks to establish our formulation, and emphasize that bridge models are suitable for image-to-video generation, which is a typical *data-to-data* generation task. Liu et al. (2023) and Chen et al. (2023c) apply bridge models to image-to-image translation and text-to-speech synthesis tasks respectively. Compared with their works, we focus on I2V tasks, building our bridge-based framework by utilizing the *frames-to-frames* essence and presenting two innovative techniques for two scenarios of training I2V models, namely fine-tuning from pre-trained text-to-video diffusion models and training from scratch.

## C  EXPERIMENT DETAILS

We provide descriptions of the datasets and metrics used in our experiments, along with implementation details for different I2V models.

### C.1  DATASETS

**UCF-101** is an open-sourced video dataset consisting of 13320 videos clips, and each video clip are categorized into one of the 101 action classes. There are three official train-test split, each of which divide the whole dataset into 9537 training video clips and 3783 test video clips. We use the whole dataset as the training data for I2V models trained from scratch on UCF-101, and use the test set to evaluate zero-shot metrics for models fine-tuned on WebVid-2M. When we evaluate zero-shot metrics on UCF-101 for text-conditional I2V models, we use the class label as the input text prompt.

**WebVid-2M** is an open-sourced dataset consisting of about 2.5 million video-text pairs, which is a subset of WebVid-10M. We only use WebVid-2M as the training data when fine-tuning I2V models from T2V diffusions in Section 5.2.

**MSR-VTT** is an open-sourced dataset consisting of 10000 video-text pairs, and we only use the test set to compute zero-shot metrics for fine-tuned models.

**Preprocess of Training Data:**  For both UCF-101 and WebVid-2M dataset, we sample 16 frames from each video clip with a fixed frame stride of 3 when training. Then we resize and center-crop the video clips to $256 \times 256$ before input it to the models.

### C.2  METRICS

**Fréchet Video Distance ( Unterthiner et al. (2018); FVD)** evaluates the quality of synthesized videos by computing the perceptual distance between videos sampled from the dataset and the models. We follow the protocol used in StyleGAN-V (Skorokhodov et al., 2022) to calculate FVD. First, we sample 2048 video clips with 16 frames and frame stride of 3 from the dataset. Then, we generate 2048 videos from the I2V models. All videos are resized to 256 $\times$ 256 before calculating FVD except for ExtDM. (ExtDM generate videos with resolution $64 \times 64$, so we compute FVD on this resolution.) After that, we extract features of those videos with the same I3D model used in the repository of StyleGAN-V [7] and calculate the Fréchet Distance.

**Inception Score (Saito et al. (2017); IS)** also evaluates the quality of the generated videos. However, computing IS need a pre-trained classifier and we only apply this metric on UCF-101. When computing IS, we use the open-sourced evaluation code and pre-trained classifier for videos from the repository of StyleGAN-V.

**CLIPSIM (Wu et al., 2021)** evaluates the consistency between video frames and the text prompt by computing the average CLIP similarity score between each frame and the prompt. We use the VIT-B/32 CLIP model (Radford et al., 2021) when evaluating zero-shot metrics on MSR-VTT.

**PIC** is a metric used by Xing et al. (2023) to evaluate the consistency of video frames and the given image by the computing average Dreamsim (Fu et al., 2023) distance between generated frames and the image condition.

### C.3  IMPLEMENTATION OF FRAMEBRIDGE AND OTHER BASELINES

We offer the implementation details of I2V models which are fine-tuned on WebVid-2M or trained from scratch on UCF-101.

#### C.3.1  FRAMEBRIDGE

**Fine-tuning on WebVid2M**  We reference the codebase of Dynamicrafter[8] to fine-tune FrameBridge, and initialize our model from the pre-trained VideoCrafter1 (Chen et al., 2023a) checkpoint. For the schedule of bridge, we adopt

---

[7]https://github.com/universome/stylegan-v

[8]https://github.com/Doubiiu/DynamiCrafter

the Bridge-gmax schedule of (Chen et al., 2023c), where $f(t) = 0$, $g(t)^2 = \beta_0 + t(\beta_1 - \beta_0)$, $\alpha_t = 1$, $\sigma_t^2 = \frac{1}{2}(\beta_1 - \beta_0)t^2 + \beta_0 t$ with $\beta_0 = 0.01$, $\beta_1 = 50$. For both FrameBridge with and without SAF technique, we fine-tune the models $\epsilon^{\hat{\Psi}}$ for 20k iterations with batch size 64. We use the AdamW optimizer with learning rate $1 \times 10^{-5}$ and mixed precision of BFloat16. We do not apply ema to the model weight during fine-tuning. The conditions $c$ and $z^i$ are incorporated into the network in the same way as DynamiCrafter, and we concatenate $\mathbf{z}_t$ with $z^i$ along the channel axis to condition the network on the prior. As the schedule $\{\alpha_t, \sigma_t\}_{t \in [0,T]}$ is different from that of the pre-trained diffusion models, we use the generalized SAF (Proposition 2).

**Training From Scratch on UCF-101** We reference the codebase of Latte[9] to train FrameBridge from scratch on UCF-101. We adopt Latte-S/2 as our bridge model with the same schedule as above and train FrameBridge for 400k iterations with batch size 40. For FrameBridge with neural prior, we also implement $F_\eta(z^i, c)$ with Latte-S/2 except that the conditioning of timestep $t$ is removed from the model. To match $z^i$ with the input shape of Latte, we replicate $z^i$ for $L$ times and concatenate them along temporal axis. We train $F_\eta(z^i, c)$ for 400k iterations with batch size 32 before training bridge models if the neural prior is applied. For both the training of bridge models and $F_\eta(z^i, c)$, we use the AdamW optimizer with learning rate $1 \times 10^{-5}$ and ema is not applied. The conditions $c$ are incorporated into the network in the same way as Latte. Since Latte-S/2 is a transformer-based diffusion network, we incorporate the condition $z^i$ by concatenate it with video latent $\mathbf{z}_t$ in the token sequence. To condition the network on prior $\mathbf{z}^i$ or $F_\eta(z^i, c)$, we concatenate them with $\mathbf{z}_t$ along the channel axis.

### C.3.2 BASELINES FOR CLASS-CONDITIONAL I2V GENERATION

**ExtDM (Zhang et al., 2024b)** is a diffusion-based video prediction model, which is trained to predict the following $m$ frames with the given first $n$ frames of a video clip. We train ExtDM with their official implementation[10] and set $n = 1, m = 15$ for our I2V setting on UCF-101.

**VDT-I2V** is our implementation of the I2V method proposed by Lu et al. (2023). They use a transformer-based diffusion network for I2V generation by directly concatenating the image condition with the token sequence of the noisy video latent $\mathbf{z}_t$. We also implement their I2V method on a Latte-S/2 model considering the similarities among transformer-based diffusion models.

### C.3.3 BASELINES FOR TEXT-CONDITIONAL I2V GENERATION

For DynamiCrafter (Xing et al., 2023), SVD (Blattmann et al., 2023), SEINE (Chen et al., 2023b), ConsistI2V (Ren et al., 2024) and SparseCtrl (Guo et al., 2025), we use the official model checkpoints and sampling code to sample videos for evaluation.

### C.3.4 ABLATION STUDIES ON NEURAL PRIOR

In Section 5.4, we ablate on the neural prior technique by comparing the performance of four models:

- **VDT-I2V**: The same model as our diffusion baseline on UCF-101.

- **VDT-I2V with neural prior as the network condition**: The same model as VDT-I2V except that we additionally condition the network on $F_\eta(z^i, c)$.

- **FrameBridge without neural prior**: A FrameBridge model implemented by utilizing the replicated image $\mathbf{z}^i$ as the prior.

- **FrameBridge with neural prior only as the network condition**: A FrameBridge model implemented by utilizing $\mathbf{z}^i$ as the prior. However, we condition the bridge model on $F_\eta(z^i, c)$ by additionally feeding it into the network through concatenation with $\mathbf{z}_t$ along the channel axis.

- **FrameBridge-NP**: A FrameBridge model implemented by utilizing $F_\eta(z^i, c)$ as the prior.

---

[9]https://github.com/Vchitect/Latte
[10]https://github.com/nku-zhichengzhang/ExtDM

---
**Algorithm 1:** Training algorithms for FrameBridge.

---

**Result:** Trained FrameBridge model $\boldsymbol{\epsilon}_{\theta}^{\hat{\Psi}}(\mathbf{z}_t, t, \mathbf{z}_T, z^i, c)$.

Set bridge process $\{\alpha_t, \sigma_t, a_t, b_t, c_t\}_{t=0}^{T}$ ;

**if** *Neural prior is used* **then**
    Train a neural prior model $F_{\eta}(z^i, c)$ with eq. (8) before training FrameBridge ;
**end**

**if** *Fine-tuned from pre-trained diffuion model* $\boldsymbol{\epsilon}_{\phi}(\mathbf{z}_t, t, c)$ **then**
    **if** *SAF is used* **then**
        Re-parameterize the input of $\boldsymbol{\epsilon}_{\theta}^{\hat{\Psi}}(\mathbf{z}_t, t, \mathbf{z}_T, z^i, c)$ by $\boldsymbol{\epsilon}_{\theta}^{\hat{\Psi}}(\mathbf{z}_t, t, \mathbf{z}_T, z^i, c) \triangleq \boldsymbol{\epsilon}_{\theta,align}^{\hat{\Psi}}(\tilde{\mathbf{z}}_t, \tilde{t}, \mathbf{z}_T, z^i, c)$ with
        eq. (24) ;
        Initialize $\boldsymbol{\epsilon}_{\theta,align}^{\hat{\Psi}}$ with the weight of $\boldsymbol{\epsilon}_{\phi}(\mathbf{z}_t, t, c)$ ;
    **else**
        Initialize $\boldsymbol{\epsilon}_{\theta}^{\hat{\Psi}}$ with the weight of $\boldsymbol{\epsilon}_{\phi}(\mathbf{z}_t, t, c)$ ;
    **end**
**else**
    Randomly initialize $\boldsymbol{\epsilon}_{\theta}^{\hat{\Psi}}(\mathbf{z}_t, t, \mathbf{z}_T, z^i, c)$;
**end**

**while** *Not reach the training budget* **do**
    Sample data $(\mathbf{z}_0, c) \sim p_{data}(\mathbf{z}_0, c)$, timestep $t$ and $\mathbf{z}_t \sim p_{bridge,t}(\mathbf{z}_t|\mathbf{z}_0, \mathbf{z}_T)$;
    Take the first frame of $\mathbf{z}_0$ as the image condition $z^i$ ;
    **if** *Neural prior is used* **then**
        $\mathbf{z}_T \leftarrow F_{\eta}(z^i, c)$ ;
    **else**
        Construct $\mathbf{z}_T$ by replicating $z^i$ ;
    **end**
    $l(\theta) = \left\| \boldsymbol{\epsilon}_{\theta}^{\hat{\Psi}}(\mathbf{z}_t, t, \mathbf{z}_T, z^i, c) - \frac{\mathbf{z}_t - \alpha_t \mathbf{z}_0}{\sigma_t} \right\|^2$ ;
    Update $\theta$ with the optimizer and loss function $l(\theta)$ ;
**end**

---

## D  MORE DISCUSSIONS ABOUT EXPERIMENT RESULTS

In this section, we provide further discussions and analysis of the results provided in Section 5.

### D.1  DYNAMIC DEGREE OF GENERATED VIDEOS

As shown by (Zhao et al., 2024), there is usually a trade-off between dynmaic motion and condition alignment for I2V models, and the high dynamic degree scores of some baseline models in Table 2 are at the cost of condition and temporal consistency. FrameBridge can reach a balance demonstrated by the multi-dimensional evaluation on VBench-I2V. Moreover, as demonstrated by Table 6 the dynamic degree of FrameBridge is not low compared with other models on the official VBench-I2V Leaderboard.

### D.2  CONTENT-DEBIASED FVD

Ge et al. (2024) points out that the FVD metric has a content bias and may misjudge the qualify of videos. As supplementary, we also provide the evaluation results of the Content-Debiased FVD (CD-FVD) on MSR-VTT in Table 7.

### D.3  LEARNING CURVE FOR VIDEO QUALITY

To illustrate the change of video quality during training, we reproduce the training process of DynamiCrafter for 20k iterations and compare the zero-shot CD-FVD metric on MSR-VTT dataset with a FrameBridge model trained during

Table 6: VBench-I2V scores related to the motion of videos for different I2V models. For all the evaluation dimensions, higher score means better performance. For results marked by *, we directly use the data of VBench-I2V Leaderboard.

| Model | Dynamic Degree | Temporal Flickering | Motion Smoothness |
|---|---|---|---|
| FrameBrdige | 35.77 | **98.01** | **98.51** |
| DynamiCrafter | **38.69** | 97.03 | 97.82 |
| SEINE | 24.55 | 95.07 | 96.20 |
| SEINE-512 × 320* | 34.31 | 96.72 | 96.68 |
| SEINE-512 × 512* | 27.07 | 97.31 | 97.12 |
| ConsistI2V* | 18.62 | 97.56 | 97.38 |

Table 7: Zero-shot CD-FVD metric on MSR-VTT dataset. We also include the FVD metric as a reference

| Model | CD-FVD ↓ | FVD ↓ |
|---|---|---|
| DynamiCrafter | 207 | 234 |
| SEINE | 420 | 245 |
| ConsistI2V | 192 | 106 |
| SparseCtrl | 454 | 311 |
| FrameBridge-100k | **148** | **95** |

the training process. As we use the same training batch size and model structure for FrameBridge and DynamiCrafter in this experiment, the training budget for two models at the same training step is also the same. As demonstrated by Figure 6, the video quality of FrameBridge is superior to that of DynamiCrafter during the training process and it also converges faster than its diffusion counterpart (*i.e.*, DynamiCrafter).

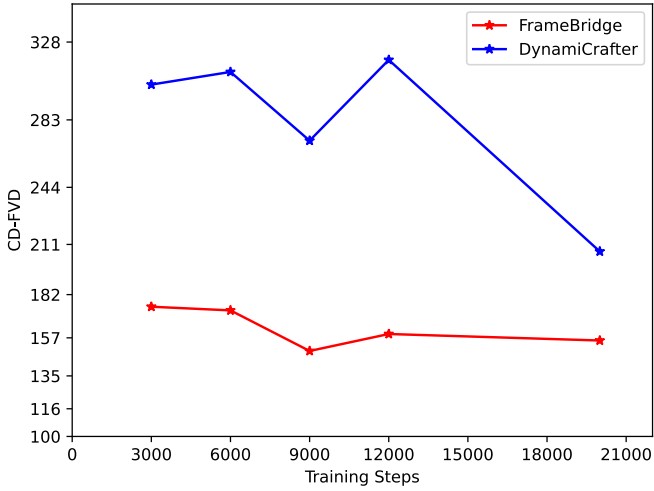

Figure 6: The learning curve of FrameBridge and DynamiCrafter.

## D.4 SAMPLING EFFICIENCY OF FRAMEBRIDGE

Since sampling efficiency is also important for I2V models, we also conduct experiments to show the quality of videos sampled with different number of sampling timesteps and compare it with DynamiCrafter and SEINE. Figure D.4 shows that the quality of videos sampled by FrameBridge is better than that of DynamiCrafter and SEINE with different timesteps (*i.e.*, 250, 100, 50, 40, 20). Moreover, we also measure the actual execution time of the sampling algorithm and show the result in Figure D.4. As illustrated by these two figures, FrameBridge can achieve good balance between sample efficiency and video quality, and there is no significant degradation in video quality when decreasing the sampling timestep from 250 to 50 or even smaller.

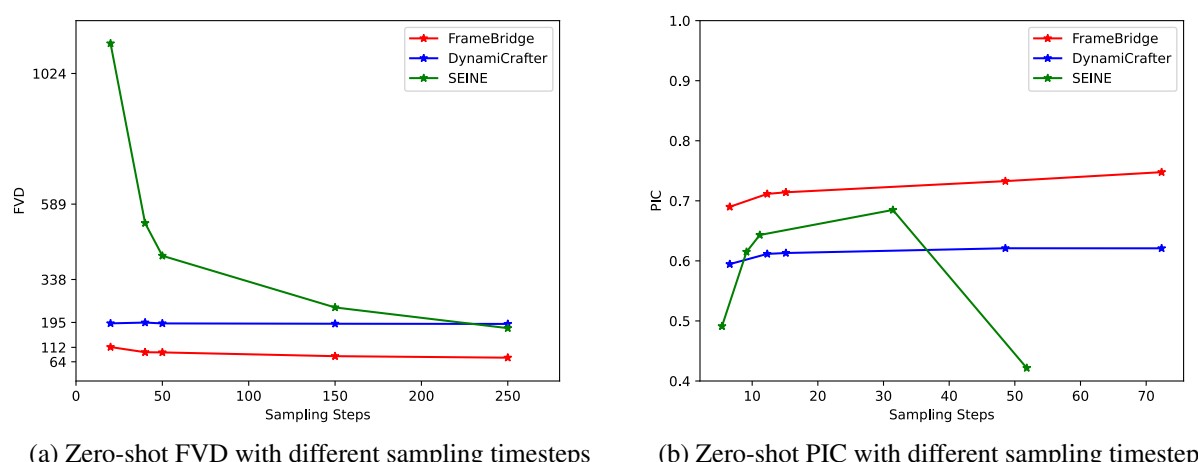

(a) Zero-shot FVD with different sampling timesteps   (b) Zero-shot PIC with different sampling timesteps

Figure 7: Video quality sampled with different number of timesteps.

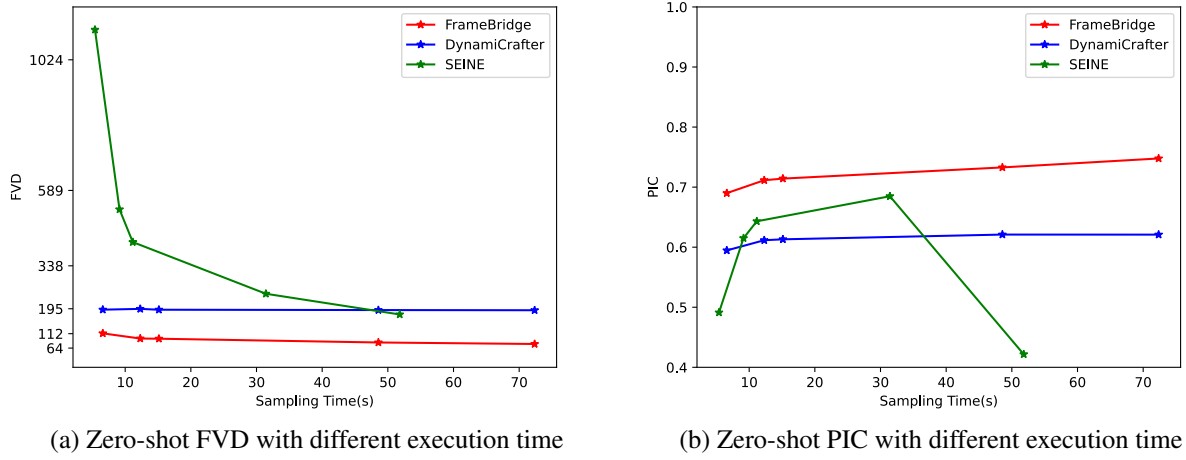

(a) Zero-shot FVD with different execution time   (b) Zero-shot PIC with different execution time

Figure 8: Video quality sampled with different execution time.

## E   PSEUDO CODE FOR THE TRAINING AND SAMPLING OF FRAMEBRIDGE

We provide the pseudo code for the training and sampling process of FrameBridge (See Algorithm 1 and 3). Meanwhile, we also provide that of diffusion-based I2V models (See Algorithm 2 and 4) to show the distinctions between FrameBridge and diffusion-based I2V models.

---

**Algorithm 2:** Training algorithms for I2V diffusion models.

---

**Result:** Trained I2V diffusion model $\epsilon_\theta(\mathbf{z}_t, t, z^i, c)$.

Set diffusion process $\{\alpha_t, \sigma_t\}_{t=0}^T$ ;

**if** *Fine-tuned from pre-trained diffuion model $\epsilon_\phi(\mathbf{z}_t, t, c)$* **then**
  | Initialize $\epsilon_\theta$ with the weight of $\epsilon_\phi(\mathbf{z}_t, t, c)$ ;
**else**
  | Randomly initialize $\epsilon_\theta(\mathbf{z}_t, t, z^i, c)$;
**end**

**while** *Not reach the training budget* **do**
  | Sample data $(\mathbf{z}_0, c) \sim p_{data}(\mathbf{z}_0, c)$, timestep $t$ and $\mathbf{z}_t = \alpha_t \mathbf{z}_0 + \sigma_t \epsilon$, where $\epsilon \sim \mathcal{N}(0, I)$ ;
  | Take the first frame of $\mathbf{z}_0$ as the image condition $z^i$ ;
  | $l(\theta) = \left\| \epsilon_\theta(\mathbf{z}_t, t, z^i, c) - \epsilon \right\|^2$ ;
  | Update $\theta$ with the optimizer and loss function $l(\theta)$ ;
**end**

---

---

**Algorithm 3:** Sampling algorithms for FrameBridge.

---

**Result:** Video latent $\mathbf{z}_0$.

Prepare a trained FrameBridge model $\epsilon_\theta^{\hat{\Psi}}(\mathbf{z}_t, t, \mathbf{z}_T, z^i, c)$ and timestep schedule $0 = t_0 < t_1 < ... < t_N = T$;

Obtain the given input image $z^i$ and additional conditions $c$ ;

**if** *Neural prior is used* **then**
  | $\mathbf{z}_T \leftarrow F_\eta(z^i, c)$ ;
  | (Here $F_\eta$ should be the same neural prior model used in the training process.)
**else**
  | Construct $\mathbf{z}_T$ by replicating $z^i$ ;
**end**

**for** $k = N$ *downto* $1$ **do**
  | Calculate the score function of bridge process $\nabla_{\mathbf{z}} \log p_{bridge,t_k}(\mathbf{z}|\mathbf{z}_T, z^i, c)|_{\mathbf{z}=\mathbf{z}_{t_k}}$ with $\epsilon_\theta^{\hat{\Psi}}(\mathbf{z}_{t_k}, t_k, \mathbf{z}_T, z^i, c)$ ;
  | Utilize a SDE solver to solve the backward bridge SDE
  | $\mathrm{d}\mathbf{z}_t = \left[ \boldsymbol{f}(t)\mathbf{z}_t - g(t)^2(\boldsymbol{s}(\mathbf{z}_t, t, \mathbf{z}_T, z^i, c) - \boldsymbol{h}(\mathbf{z}_t, t, \mathbf{z}_T, z^i, c)) \right] \mathrm{d}t + g(t)\mathrm{d}\bar{\boldsymbol{w}}$ from $\mathbf{z}(t_k) = \mathbf{z}_{t_k}$ to obtain
  | $\mathbf{z}_{t_{k-1}}$ ;
**end**

Return $\mathbf{z}_0$ ;

---

---

**Algorithm 4:** Sampling algorithms for I2V diffusion models.

---

**Result:** Video latent $\mathbf{z}_0$.

Prepare a trained I2V diffusion model $\epsilon_\theta(\mathbf{z}_t, t, z^i, c)$ and timestep schedule $0 = t_0 < t_1 < ... < t_N = T$;

Obtain the given input image $z^i$ and additional conditions $c$ ;

Sample a latent $\mathbf{z}_T \sim \mathcal{N}(0, \sigma_T^2 I)$ ;

**for** $k = N$ *downto* $1$ **do**
  | Calculate the score function of diffusion process $\nabla_{\mathbf{z}} \log p_{diff,t_k}(\mathbf{z}|z^i, c)|_{\mathbf{z}=\mathbf{z}_{t_k}}$ with $\epsilon(\mathbf{z}_{t_k}, t_k, z^i, c)$ ;
  | Utilize a SDE solver to solve the backward diffusion SDE
  | $\mathrm{d}\mathbf{z}_t = \left[ \boldsymbol{f}(t)\mathbf{z}_t - g(t)^2 \nabla_{\mathbf{z}_t} \log p_{t,diff}(\mathbf{z}_t|z^i, c) \right] \mathrm{d}t + g(t)\mathrm{d}\bar{\mathbf{w}}$ from $\mathbf{z}(t_k) = \mathbf{z}_{t_k}$ to obtain $\mathbf{z}_{t_{k-1}}$ ;
**end**

Return $\mathbf{z}_0$ ;

---

## F  MORE QUALITATIVE RESULTS

We show several randomly selected samples of FrameBridge below, and more synthesized samples can be visited at:
`https://framebridgei2v.github.io/`

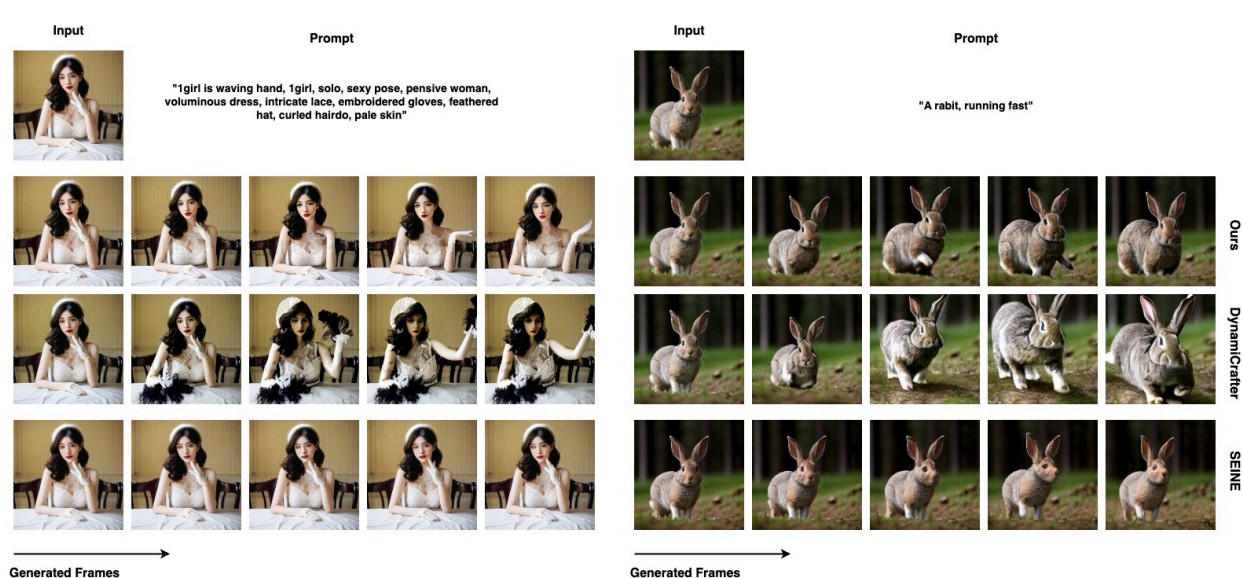

Figure 9: **Qualitative comparisons between FrameBridge and other baselines.** FrameBridge outperforms other diffusion-based methods in appearance consistency and video quality.

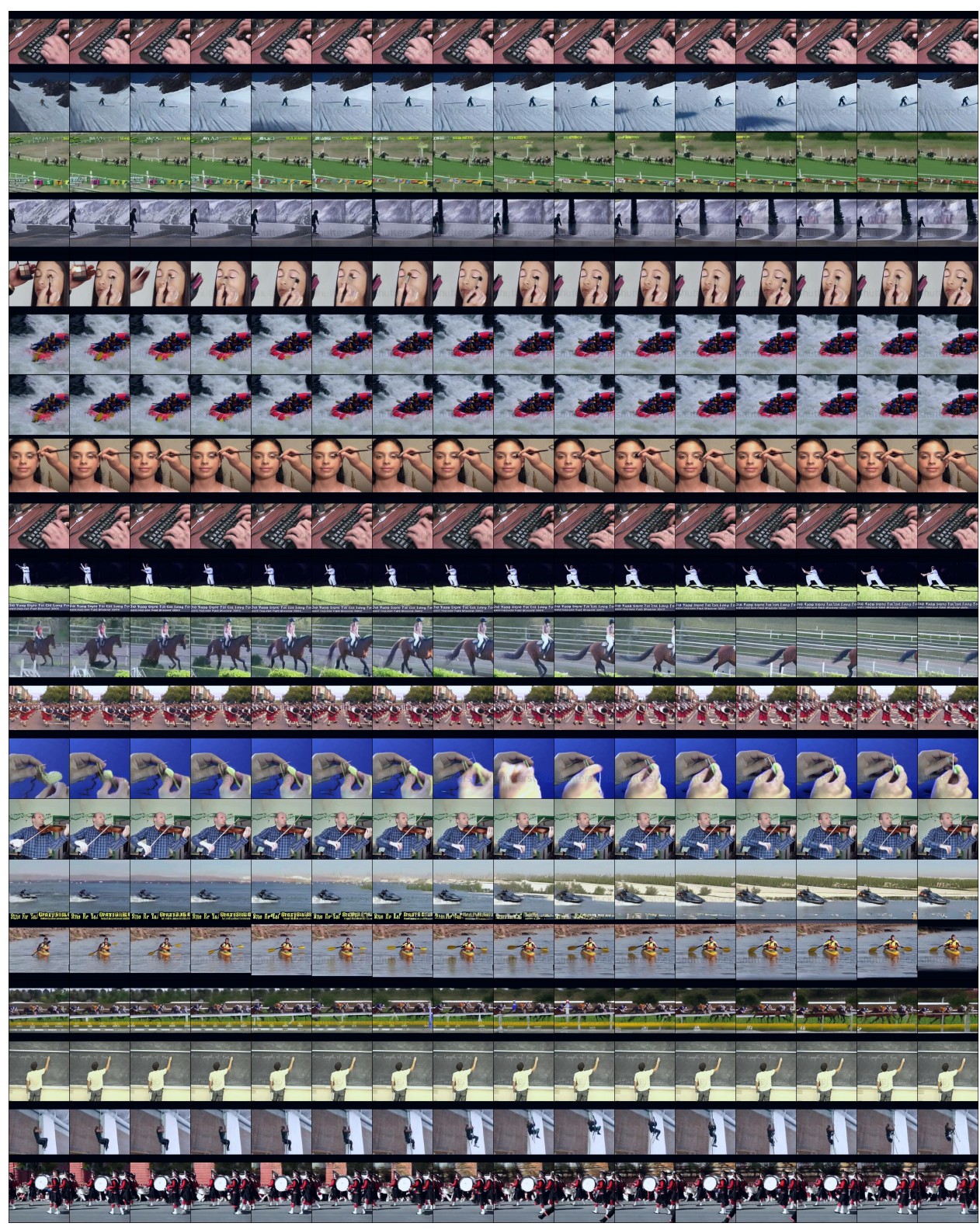

27

Figure 10: Zero-shot generation results of fine-tuned FrameBridge (with SAF) on UCF-101.

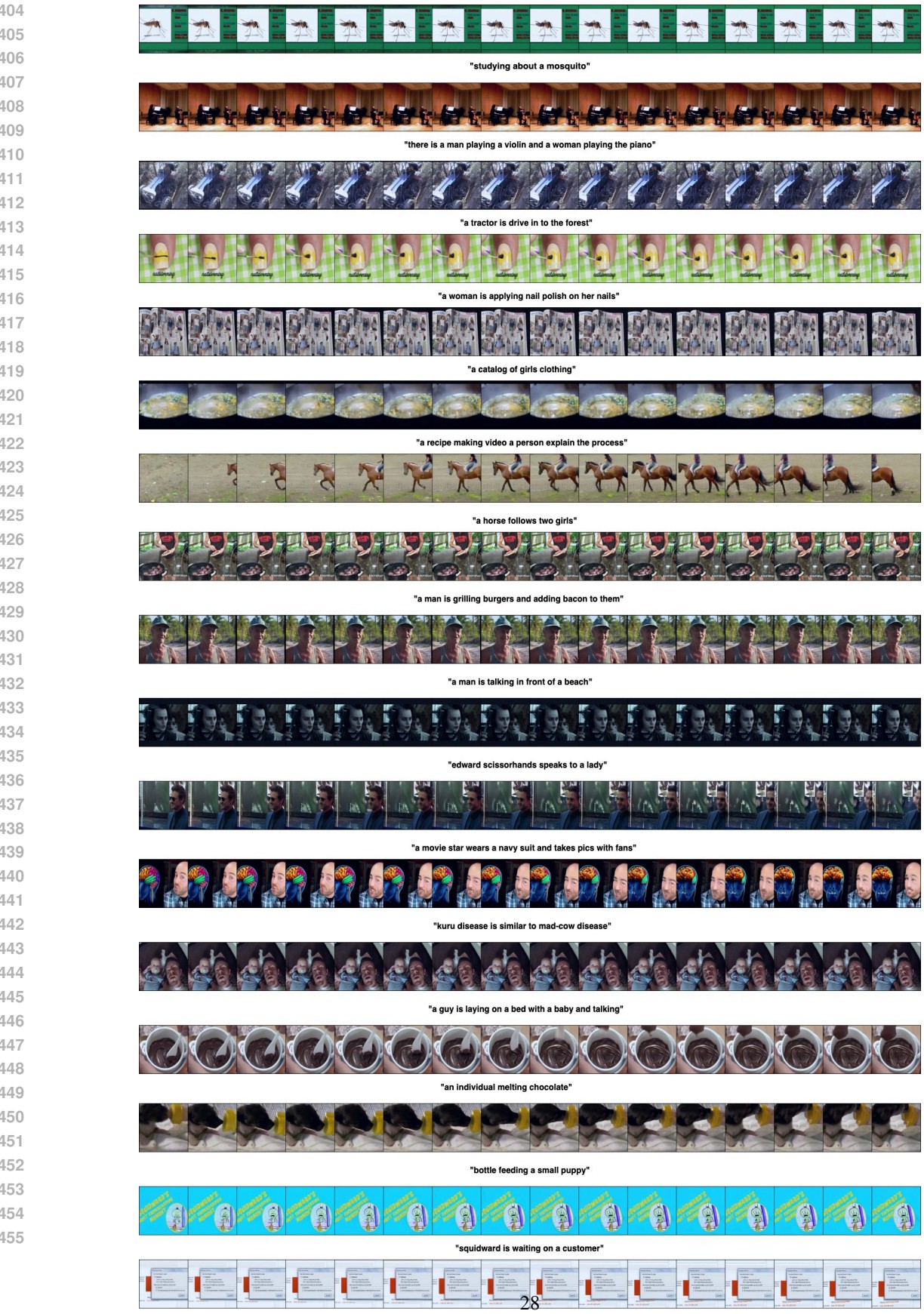

Figure 11: Zero-shot generation results of fine-tuned FrameBridge (with SAF) on MSR-VTT.

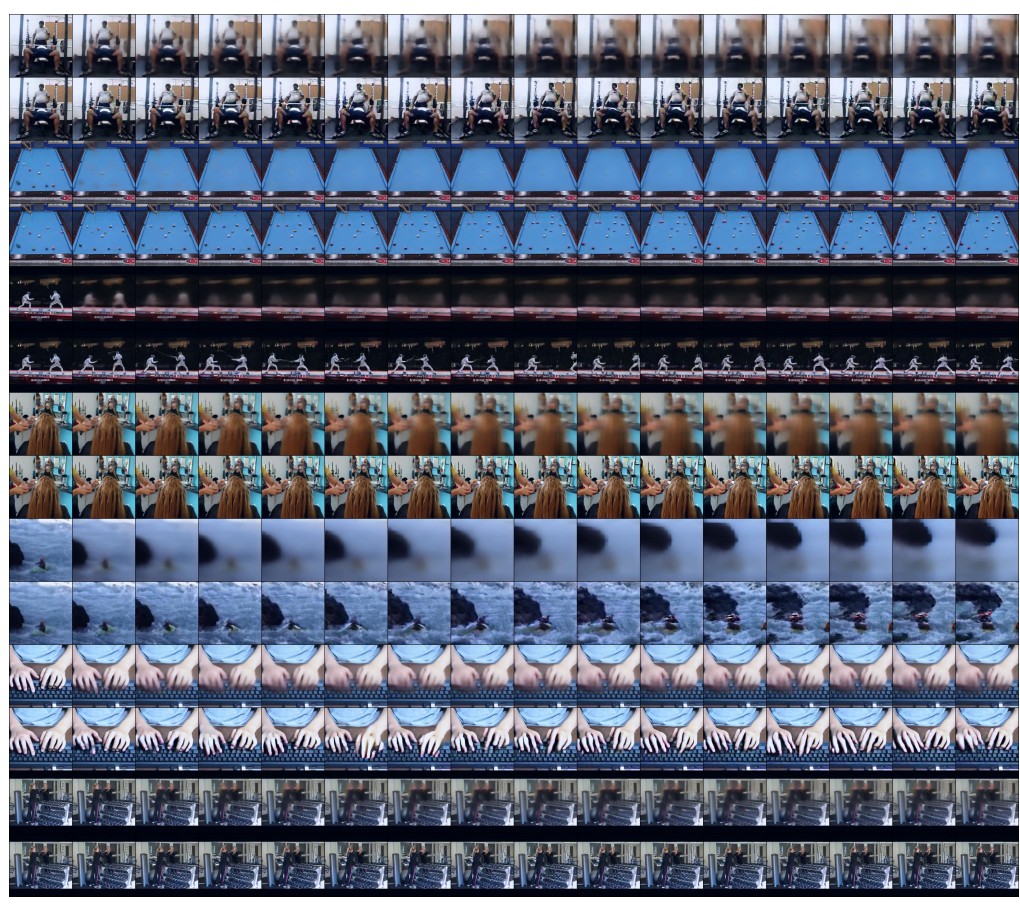

Figure 12: Non-zero-shot generation results of FrameBridge-NP on UCF-101. We use two lines to present a neural prior and the corresponding generated video.

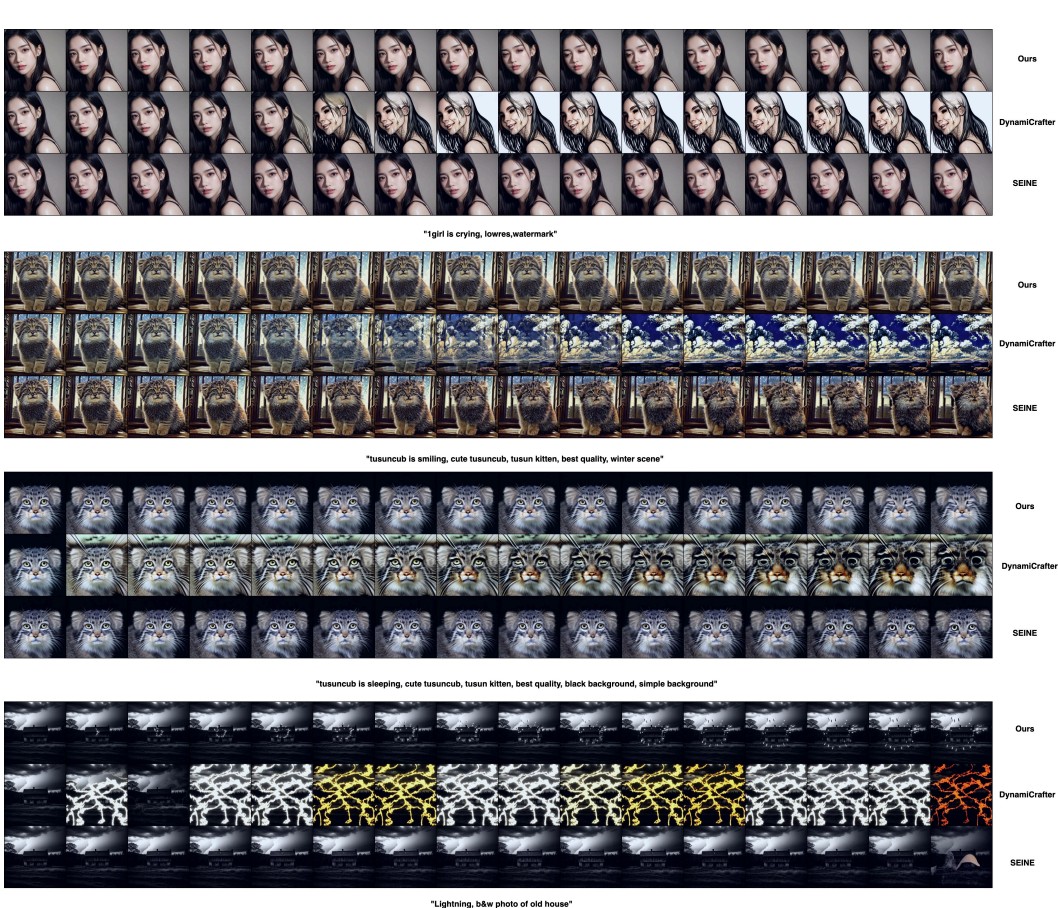

Figure 13: Comparisons between fine-tuned FrameBridge and other diffusion-based I2V models.

