# OpenReview forum: "FrameBridge: Improving Image-to-Video Generation with Bridge Models"
_ICLR.cc/2025/Conference — Submitted to ICLR 2025_

### Official Review · Reviewer_ScY5 · 2024-10-15

**Soundness:** 3
**Presentation:** 2
**Contribution:** 2
**Rating:** 5
**Confidence:** 3

**Summary:**

The paper introduces an interesting image-to-video diffusion-based methods by leveraging the Denoising Diffusion Bridge Models (DDBM). It sets the initial state of denoising as temporally replicated first frame (i.e., static videos) and uses the DDBM to connect the distributions of the static videos and real videos sharing the same first frame. The reported results is better than naive diffusion models.

**Strengths:**

- The overall story is smooth and sounds interesting.

- The discussion on related works is sufficient.

- The experimental results seems to support the claim of contributions in the paper.

**Weaknesses:**

1. **I have a fundamental concern that I think the proposed method has no inherent difference between naive video diffusion models, especially considering the re-parameterization trick proposed in the paper.**

The paper says "Formulating a frame-to-frames generation task with a conditional noise-to-data sampling process, diffusion-based I2V systems suffer the difficulties to generate high-quality samples from uninformative Gaussian noise". This is to say that the authors believe the model can hardly learn to predict good videos $x_0$ from the mixture state of noise and data, that is $x_t = \alpha_t x_0 + \sigma_t \epsilon$.  Instead, providing the model the information from the statics images can help the model learn. The input for bridge models is formulated as $x_t = a_t x_0 + b_t \epsilon + c_t x_T$, where $x_T$ is the replicated initial frames.  This is also the reasonability of the bridge models.

However, as proposed in Equation 7 of the paper, the authors remove all the information $x_T$ (The information of the initial frame is all removed), and the noisy state $x_t$ returns to the normal forms of standard diffusion models, which is the weighted sum of the target $x_0$ and the noise $\epsilon$.  Even though the authors claim that this operation helps them to adapt the standard diffusion models into the diffusion bridge models, I think this operation breaks the basic property of diffusion bridge models.

To this end, the only difference between the proposed method and the standard diffusion model is the prediction types. For standard DDPM, the model learns to predict the $\epsilon$. For the proposed method, the model learns to predict the $\epsilon + m_t x_T$. $m_t$ can be some time-dependent factors (not strict symbol form).   Since the model's input is the same now and the predictions just have an additional term $m_t x_T$ which is known at either training and inference stages, the influence of $x_T$ will be eliminated due to the sampling starting point difference between diffusion models and bridge models. I don't think the prediction type difference can make any difference. Therefore, I am in deep concern that the proposed method might have no intrinsic difference from naive video diffusion models.

I am not very sure about my judgement though. I hope to propose the concern and inspire other experienced reviewers do further check on the  proposed technique.


2. **Connecting similar distributions is not uncommon.** Many works proposed similar ideas on image fields [1] [2] [3] for applications that they build a connection between the target image fields and the source image fields instead of simply using the source images as conditions.

3.  **Detailed FVD learning curve for different baselines.** I would like the authors provide the detailed staged FVD learning curves (or other metrics) for different baselines. They can clearly show the learning speed of different methods and show performance gap of different methods trained with the same training budget.

4. In table 4, I would like to see the FVD for gaussian prior and Fn conditions. Since we can see from the table that, when both using Fn as conditions, the model have very similar performance with replicate prior and neural prior. This might indicate that the condition is more important than prior.

5. It is known that FVD has a very strong content bias and tends to misjudge the quality of video motion. Therefore I want the authors report the new-FVD proposed in the previous work. [4]



[1]  FMBoost: Boosting Latent Diffusion with Flow Matching. ECCV 2024 Oral

[2]  ResShift: Efficient Diffusion Model for Image Super-resolution by Residual Shifting. NeurIPS 2023, Spotlight, TPAMI@2024.

[3] Residual Denoising Diffusion Models. CVPR 2024.

[4] Content-Debiased FVD for Evaluating Video Generation Models.  CVPR 2024.

**Questions:**

I want the authors to show the training/inference pseudo code of their proposed method and standard video diffusion methods. It is the best way to provide a clear difference of these two methods.

---

> ### Comment · Reviewer_ScY5 · 2024-11-25
> **Do the authors still want to submit their rebuttal?**
>
> There are only about 2 days left for the review discussion. I wonder whether the authors still want to submit their rebuttal.

---

> > ### Author Response · Authors · 2024-11-26
> >
> > Yes, we are now carefully prepare the materials and will submit a rebuttal soon. As our computational resources are very limited, it takes us some time to obtain the experimental results. We are glad to engage in a detailed and thorough discussion later.

---

> ### Author Response · Authors · 2024-11-27
>
> We appreciate the reviewer for recognizing the experimental results in our work and the instructive comments. We are glad to engage in a further discussion and address the reviewer's concerns.
>
> **Reply to W1: Differences between FrameBridge and diffusion-based I2V models**
>
> We thank the reviewer for carefully reviewing our method and providing detailed feedback. We will clarify the differences between FrameBridge and diffusion-based I2V models below.
>
> At the beginning, we want to emphasize that since the sampling and training process are closely related for bridge models and diffusion models, we should take the sampling algorithm into account when discussing the two types of models. No matter if we use SAF or not, **the sampling process of FrameBridge is different from that of diffusion-based I2V models**.
>
> To be more specific, diffusion models generate data from noise with the backward diffusion SDE
>
> $$ \mathrm{d}x_t = \left[f(x_t, t) - g(t)^2 \nabla_x \log p_{diff, t}(x_t, y)\right]\mathrm{d}t + g(t) \mathrm{d}w, \quad x_T \sim \mathcal{N}(0, I), $$
>
> while the sampling process of bridge models is equivalent to solve the backward bridge SDE
>
> $$\mathrm{d}x_t = \left[\tilde{f}(x_t, t) - \tilde{g}(t)^2 \left(\frac{1}{2}\nabla_x \log p_{bridge, t}(x_t, y, x_T) - \tilde{h}(x_t, t, y)\right)\right]\mathrm{d}t + \tilde{g}(t) \mathrm{d}w, \quad x_T = y.$$
>
> Here $f, g, \tilde{f}, \tilde{g}, \tilde{h}$ can be directly calculated given the bridge and diffusion schedule, and $p_{diff, t}, p_{bridge, t}$ are the marginal distribution of forward diffusion and bridge process. Similar to diffusion models, diffusion bridge models estimate the bridge score function $\nabla_x \log p_{bridge, t}(x_t, y, x_T)$ with the help of a neural network $x_\theta(x_t, t, y, x_T)$ (we use the data-prediction bridge models [1, 2] as an example for our illustration) trained with denoising bridge score matching (DBSM) loss, and the training loss is:
>
> $$L_{bridge}(\theta) = \mathbb{E} \left[ \lVert x_\theta(x_t, t, y, x_T)- x_0\rVert^2 \right]$$
>
> SAF modifies the input parameterization of $x_\theta(\cdot)$, **but does not change the sampling process and the learning target $L_{bridge}(\theta)$ (i.e., we still train the model $x_\theta$ to estimate the bridge score function and sampling with the backward bridge SDE from a deterministic point $x_T$).** Therefore, it is not exactly that "the only difference between the proposed method and the standard diffusion model is the prediction types". The advantages of bridge models over diffusion models lie not only in training but also in the sampling process, and the latter may even be more crucial.
>
> More intuitively speaking, both FrameBridge with and without the proposed SAF technique generate videos from a **deterministic prior**, thus following a **frames-to-frames** process, which is naturally suitable for I2V generation tasks as demonstrated by our experimental results. FrameBridge with SAF only modifies the input of bridge models and **still sample videos by solving backward bridge SDE with the estimated score of bridge process**. This makes it inherently different comapred with diffusion-based I2V models.
>
> We appreciate the reviewer's suggestion to use pseudo code to illustrate the differences between bridge models and diffusion models, and we will add that in the final version of our paper later.
>
> **Reply to W2: Main contributions of our work**
>
> We want to emphasize that our main contribution is to extend bridge models to video domain (specifically, to Image-to-Video generation), and design techniques (e.g., SAF and neural prior) which **further improves bridge models.**
>
> - **SNR-Aligned Fine-tuning**: For video generation tasks, it is a widely adopted method to fine-tune a pre-trained T2V base model for other tasks. To the best of our knowledge, it is **the first trial to fine-tune a pre-trained diffusion model to a bridge model**, which is different from previous works of bridge models in image domain [2, 3].
>
> - **Neural prior:** **Bridge models for image-to-image translation tasks can directly use the condition image as the prior of bridge models, which is not the case for I2V generation.** To obtain a prior in the same Euclidean space as the target, we first use the replicated image condition and then **explore to generate the prior with another single-step regression model (i.e., neural prior).**

---

> ### Author Response · Authors · 2024-11-27
>
> **Reply to W3: FVD learning curve**
>
> As suggested by reviewer ScY5, we use the new-FVD (content-debiased FVD [4]) to evaluate the video quality generated by I2V models.
>
> To comapre FrameBridge with baseline models, we provide the new-FVD curve of DynamiCrafter and FrameBridge on MSR-VTT. (Due to the limiation of computational resources, we can not reproduce the training procedure of SEINE, and use the released model for evaluation)
>
> **Table: new-FVD on MSR-VTT at different training iterations**
> Training iterations | FrameBridge | DynamiCrafter (diffusion counterpart)
> --- | --- | ---
> 3k | 174.99 | 303.41|
> 6k | 172.87 | 310.72|
> 9k | 149.41| 271.00|
> 12k | 159.22 | 317.69|
> 20k | 155.46 | 206.94|
>
> Since we use the same model structure and batch size for the FrameBridge and DynamiCrafter model, it can be directly concluded that FrameBridge generate better videos than its diffusion counterpart during the whole training process and it converges faster.
>
> **Reply to W4: Ablation for neural prior**
>
> To address the reviewer's concern, we report the FVD for diffusion models (with gaussian prior) using $F_\eta$ as a condition:
>
> Method|  Prior | Condition on $F_\eta$ | FVD $\downarrow$
> -- | -- | -- | --
> VDT-I2V | Gaussian | N | 171
> VDT-I2V | Gaussian | Y | 132
> FrameBridge | replicated | N | 154
> FrameBridge | replicated | Y | 129
> FrameBridge-NP | neural | Y | **122**
>
> Comparing the second and last row of the table, it can be seen that bridge models can utilize the neural prior $F_\eta$ better by setting it as both the prior (i.e. the start point of sampling process) and model condition, and the prior plays an import role in bridge models.
>
> **Reply to W5: Evaluation metrics**
>
> We evaluate the new-FVD value of FrameBridge, DynamiCrafter, and SEINE on MSR-VTT:
>
> Model | New-FVD $\downarrow$ |
> --- | --- |
> FrameBridge | **155.46**
> DynamiCrafter | 206.94
> SEINE | 420.84 |
>
> Moreover, we adopt other reviewer's suggestions to provide evaluations on the advanced benchmark VBench [5] for video generation.
>
> **Table: Scores of VBench-I2V (higher the better)**
> Model     | I2V Subject Consistency | I2V Background Consistency | Temporal Flickering | Motion Smoothness | Dynamic Degree | Aesthetic Quality | Imaging Quality
> -------- | -------- | ------- | ----- | -----  | -----  | -----  | -----  |
> DynamiCrafter | 95.40 | 96.22 | 97.03 | 97.82 | **38.69** | **59.40** | 62.29
> SEINE | 93.45 | 94.21 | 95.07 | 96.20 | 24.55 | 56.55 | **70.52**
> FrameBridge  | **96.24** | **97.25** | **98.01** | **98.51** | 35.77 | 59.38 | 63.28
>
> FrameBridge outperforms other baselines on most of the dimensions, demonstrating the higher overall quality of videos generated by bridge models.
>
> **About dynamic degree:** There is usually a trade-off between dynmaic motion and condition alignment [13] for I2V models, and the high dynamic degree scores of some baseline models are at the cost of condition and temporal consistency. FrameBridge can reach a balance demonstrated by the multi-dimensional evaluation on VBench-I2V. Moreover, the motion dynamic of FrameBridge is not low compared with other models on
> [VBench-I2V leaderboard](https://huggingface.co/spaces/Vchitect/VBench_Leaderboard).
>
> **Table: Scores related to motion**
> Model     | Dynamic Degree | Temporal Flickering | Motion Smoothness
> -------- | -------- | ---- | --- |
> FrameBrdige | 35.77 | 98.01 | 98.51
> SEINE [7]-512 $\times$ 320* | 34.31 | 96.72 | 96.68
> SEINE-512 $\times$ 512* | 27.07 | 97.31 | 97.12
> ConsistI2V [8]* | 18.62 | 97.56 | 97.38
>
> *: We directly use the evaluation results provided by [VBench-I2V leaderboard](https://huggingface.co/spaces/Vchitect/VBench_Leaderboard).
>
> [1] Kaiwen Zheng*, Guande He*, Jianfei Chen, Fan Bao, and Jun Zhu. Diffusion Bridge Implicit Models. arXiv preprint arXiv:2405.15885, 2024.
>
> [2] Linqi Zhou, Aaron Lou, Samar Khanna, and Stefano Ermon. Denoising diffusion bridge models. In ICLR, 2024.
>
> [3] Guan-Horng Liu, Arash Vahdat, De-An Huang, Evangelos A Theodorou, Weili Nie, and Anima Anandkumar. I2sb: Image-to-image schrodinger bridge. In ICML, 2023.
>
> [4] Songwei Ge, Aniruddha Mahapatra, Gaurav Parmar, Jun-Yan Zhu, and Jia-Bin Huang. On the Content Bias in Fréchet Video Distance. In CVPR, 2024.
>
> [5] Ziqi Huang, Yinan He, Jiashuo Yu, Fan Zhang, Chenyang Si, Yuming Jiang, Yuanhan Zhang, Tianxing Wu, Qingyang Jin, Nattapol Chanpaisit, Yaohui Wang, Xinyuan Chen, Limin Wang, Dahua Lin, Yu Qiao, and Ziwei Liu. VBench: Comprehensive Benchmark Suite for Video Generative Models. In CVPR, 2024.

---

> ### Comment · Reviewer_ScY5 · 2024-12-02
>
> Thanks for the author's rebuttal and further experiments and discussions. I have read the comments from the reviewers and the author's responses.
>
> 1. For my first concern, it is not well addressed.  The explanation of the authors does not reduce my concern. The **re-parameterization trick proposed in the paper will influence the calculation of the score function**. Please provide **proof of the equivalence of score function** when using the re-parameterization.
>
> 2. For my requests for checking the pseudo-code of training/inference, the authors only responded with "they will". I am confused about why they do not want to show the comparison. It is the most straightforward way to do so.
>
> 3. Additionally, the comments shared by other reviewers that the visual quality of generated videos is bad also raise my concerns about the visual quality. The demo video presented in the paper and pages is not convincing. It has a large gap to many open-sourced image-to-video diffusion models. How can we expect it to work better in practice?
>
>
> Overall, I appreciate the efforts the authors have paid to the rebuttal, but considering the above concerns I maintain my rate for rejection.

---

> > ### Author Response · Authors · 2024-12-02
> >
> > Dear Reviewer ScY5,
> >
> > We sincerely appreciate your time and effort in reading our responses, and we are sorry that we do not address some of your concerns. Thanks for your feedback and indicating the aspects that need improvement in our work. (For the second one, at the time, we mentioned "we will" as it was our plan to add it to the final version of the manuscript later, and the trianing/inference pseudo code of FrameBridge and Diffusion-based I2V models are now available in Appendix E.) Nevertheless, we are very grateful for all your feedback and will strive to enhance our work in the future.

---

> > > ### Comment · Reviewer_ScY5 · 2024-12-02
> > >
> > > Sorry for not noticing the new revision in the appendix.

---

### Official Review · Reviewer_pN4E · 2024-10-28

**Soundness:** 2
**Presentation:** 3
**Contribution:** 2
**Rating:** 3
**Confidence:** 5

**Summary:**

In this paper, the authors propose FrameBridge to achieve the image-to-video generation. Unlike previous diffusion-based methods, FrameBridge starts from a deterministic data point, the static image, and then maps it to the video latent through a bridge model. To achieve that, they further propose two techniques, SNR-
Aligned Fine-tuning (SAF) and neural prior.

**Strengths:**

1. Solving the distribution gap of training and inference is a reasonable motivation.
2. The video generated by FrameBridge is smoother with less mutation.
3. Compared to previous methods, FrameBridge achieves better metric scores on WebVid-2M and UCF-101.

**Weaknesses:**

1. Insufficient reference to related work. For example, “Common Diffusion Noise Schedules and Sample Steps are Flawed” proposes to align the SNR between the training stage and the inference stage.
2. Some videos on the website can not be played. It seems that authors still modify the website when the review period starts.
3. The video quality is not satisfactory. First, Maybe due to the size of the training set, the visual quality is not competitive (the worst one in my assigned papers).
4. It seems that neural prior will limit the dynamics of the generated results. The transformation starting point of all frames comes from the transformation of the same frame. In this case, most generated results only contain small motions. The selected indicators are not helpful in judging the motion of the video. Alternatively, evaluating on a comprehensive benchmark like VBench will make the comparison more convincing.
5. Abstract: “modelling” -> “modeling”.

**Questions:**

Can you show the best ten cases from FrameBridge? In the field of video generation, the upper bound of the model is also a very important reference indicator.

---

> ### Author Response · Authors · 2024-11-27
>
> We appreciate the reviewer's recognition of the motivation and underlying ideas behind our approach. To address the reviewer's concerns, we provide more experimental results and clarifications of our method below.
>
> **Reply to W1: Discussions about related works**
>
> [1] is one of the representative works to manipulate diffusion schedules in the inference stage, with the attempt to align the marginal distributions between training and inference stages. In the field of video diffusion models, [2, 3] share the same underlying philosophy and use a noisy latent related to image conditions as the starting point of sampling process. FrameBridge has two main distinctions compared to these works:
>
>  1. [1, 2, 3] are all diffusion models, and the starting point of sampling process is **a Gaussian distribution with certain variance scale**. However, FrameBridge follows the framework of bridge models [4] and the sampling process begins with a **deterministic latent $z_T$**.
>
>  2. [1, 2, 3] focus on the sampling process of a diffusion model and do not modify the training process. These methods are applied to already trained diffusion models. In contrast, FrameBridge **trains an I2V bridge model with the denoising bridge score matching loss (DBSM)** [4, 5], which is fine-tuned from a T2V diffusion model or trained from scratch and outperforms its diffusion counterparts trained with the same method.
>
> Meanwhile, we also want to clarify that SAF aims to **align the marginal distribution of input latents in the fine-tuning stage with that of the pre-trained diffusion model**. The objective of SAF is to effectively fine-tune a pre-trained diffusion model to a bridge model. Specifically, in I2V generation tasks, it fine-tunes a T2V diffusion model to an I2V bridge model.
>
> We hope the above clarification can address the reviewer's concerns and we are willing to have a further discussion.

---

> ### Author Response · Authors · 2024-11-27
>
> **Reply to W3: Video quality of FrameBridge**
>
> In the first version of our paper, constrained by the very limited computational resources, we fine-tune a FrameBridge model from videocrafter-256 [6] for **20k iterations** and compare it with a DynamiCrafter-256 [7] model **trained with the same budget**, aiming to demonstrate the advantage of I2V bridge models over their diffusion counterparts. To address the reviewer's concerns, we train a FrameBridge model for longer time (**100k iterations**) and compare it with baselines. Adopting the suggestion of reviewer pN4E, we provide the evaluation results on VBench-I2V [8], which is widely acknowledged as a reliable and comprehensive benchmark for video generation.
>
> **Table: Scores of VBench-I2V (higher the better)**
> Model     | I2V Subject Consistency | I2V Background Consistency | Temporal Flickering | Motion Smoothness | Dynamic Degree | Aesthetic Quality | Imaging Quality
> -------- | -------- | ------- | ----- | -----  | -----  | -----  | -----  |
> DynamiCrafter (released checkpoint in [7]) | 95.40 | 96.22 | 97.03 | 97.82 | 38.69 | **59.40** | 62.29
> SEINE [9] | 93.45 | 94.21 | 95.07 | 96.20 | 24.55 | 56.55 | **70.52**
> SparseCtrl [10] | 88.39 | 92.46 | 91.78 | 94.25 | **81.95** | 49.88 | 69.35
> FrameBridge  | **96.24** | **97.25** | **98.01** | **98.51** | 35.77 | 59.38 | 63.28
>
> The above table shows that the overall video quality of FrameBridge are higher than the baselines.
>
> **About dynamic degree**
>
> There is usually a trade-off between dynmaic motion and condition alignment [11] for I2V models, and the high dynamic degree scores of some baseline models are at the cost of condition and temporal consistency. FrameBridge can reach a balance demonstrated by the multi-dimensional evaluation on VBench-I2V. Moreover, the motion dynamic of FrameBridge is not low compared with other models on
> [VBench-I2V leaderboard](https://huggingface.co/spaces/Vchitect/VBench_Leaderboard):
>
> **Table: Scores related to motion**
> Model     | Dynamic Degree | Temporal Flickering | Motion Smoothness
> -------- | -------- | ---- | --- |
> FrameBrdige | **35.77** | **98.01** | **98.51**
> SEINE [7]-512 $\times$ 320* | 34.31 | 96.72 | 96.68
> SEINE-512 $\times$ 512* | 27.07 | 97.31 | 97.12
> ConsistI2V [8]* | 18.62 | 97.56 | 97.38
>
> *: We directly use the evaluation results provided by [VBench-I2V leaderboard](https://huggingface.co/spaces/Vchitect/VBench_Leaderboard).
>
> We would like to clarify that the training budget, the dataset size, the number of model paramters and the video resolution are all crucial factors for the quality of generated videos. Due to the limitation of computational resources, currently we can only train our models at 256 $\times$ 256 resolution with certain model size and training iterations. Moreover, the training dataset of some other models may consist of closed-source datasets and we only have access to the open-source ones. Nevertheless, we have made an effort to train an I2V bridge model under these constraints which outperforms the diffusion counterparts. **The main target and contribution of our work is not to train a SOTA I2V diffusion model, but to propose a bridge-based I2V method and demonstrate the effectiveness with resonable and fair experimental results.** We hope the evaluation results of both traditional metrics (FVD, IS, CLIPSIM, PIC) and comprehensive video benchmark (VBench-I2V) can address the concerns of the reviewer.

---

> ### Author Response · Authors · 2024-11-27
>
> **Reply to W4: Dynamic degree of FrameBridge and FrameBridge-NP**
>
> We would like to gently point out that "the transformation starting point of all frames comes from the transformation of the same frame" is a misunderstanding of our method. The above evaluation results has shown that FrameBridge will not limit the dynamic degree of generated videos, we will have a further discussion about that.
>
> **Theoretically, I2V bridge models will not limit the motion of the generated videos.** Formally speaking, the latent of a bridge process at timestep $t$ (i.e., $z_t$) has the same marginal distribution as $a_t z_0 + b_t z_T + c_t \epsilon$, where
> $$a_t = \alpha_t (1 - \frac{SNR_T}{SNR_t}), b_t = \frac{\alpha_t}{\alpha_T}\frac{SNR_T}{SNR_t}, c_t = \sigma_t \sqrt{1 - \frac{SNR_T}{SNR_t}}$$
> (Here $\alpha_t, \sigma_t$ is the schedule of a diffusion process used to construct the bridge process [4], and $SNR_t = \frac{\alpha_t^2}{\sigma_t^2}$.) Although at timestep $T$, $a_T = 0, b_T = 1, c_T = 0$, and $z_T$ is a deterministic point, the coefficient $c_t > 0$ at $t \in (0, T)$. Intuitively, different from the sampling process of a diffusion model (where the noise scale is monotonically decreasing), the noise scale of a bridge latent $z_t$ will first increase to a high point and then decrease during the whole sampling process. **Notably, the sampling process of a bridge model is also a iterative process, instead of a one-step transformation.**
>
> Moreover, when neural prior is not applied, the latent $z_T$ is obtained by replicating the first frame. If we use neural prior, $z_T$ will be replaced by a regression model $F_\eta(z^i, c)$ ($z^i$ is the first frame and $c$ is additional conditions), it will definitely contain more information about motion than simply replicating $z^i$. So, neural prior do not limit the dynamics of generated videos and will add to the dynamics on the contrary.
>
> **Reply to W2 and Question**
>
> We will update our project page later and show more cases generated by a FrameBridge model trained longer (i.e. 100k iterations).
>
> Meanwhile, we thank the reviewer to point out the typo and will carefully check our writing. We will upload our revised version of the paper later.
>
> [1] Shanchuan Lin, Bingchen Liu, Jiashi Li, and Xiao Yang. Common Diffusion Noise Schedules and Sample Steps are Flawed. In WACV, 2024.
>
> [2] Tianxing Wu, Chenyang Si, Yuming Jiang, Ziqi Huang, and Ziwei Liu. Freeinit: Bridging initialization gap in video diffusion models. arXiv preprint arXiv:2312.07537, 2023.
>
> [3] Weiming Ren, Huan Yang, Ge Zhang, Cong Wei, Xinrun Du, Wenhao Huang, and Wenhu Chen. ConsistI2V: Enhancing Visual Consistency for Image-to-Video Generation. arXiv preprint arXiv:2402.04324, 2024.
>
> [4] Linqi Zhou, Aaron Lou, Samar Khanna, and Stefano Ermon. Denoising diffusion bridge models. In ICLR, 2024.
>
> [5] Zehua Chen, Guande He, Kaiwen Zheng, Xu Tan, and Jun Zhu. Schrodinger Bridges Beat Diffusion Models on Text-to-Speech Synthesis. arXiv preprint arXiv:2312.03491, 2023.
>
> [6] Haoxin Chen, Menghan Xia, Yingqing He, Yong Zhang, Xiaodong Cun, Shaoshu Yang, Jinbo Xing, Yaofang Liu, Qifeng Chen, Xintao Wang, Chao Weng, Ying Shan. VideoCrafter1: Open diffusion models for highquality video generation. arXiv preprint arXiv:2310.19512, 2023.
>
> [7] Jinbo Xing, Menghan Xia, Yong Zhang, Haoxin Chen, Wangbo Yu, Hanyuan Liu, Gongye Liu, Xintao Wang, Ying Shan, and Tien-Tsin Wong. DynamiCrafter: Animating Open-Domain Images with Video Diffusion Priors. In ECCV, 2024.
>
> [8] Ziqi Huang, Yinan He, Jiashuo Yu, Fan Zhang, Chenyang Si, Yuming Jiang, Yuanhan Zhang, Tianxing Wu, Qingyang Jin, Nattapol Chanpaisit, Yaohui Wang, Xinyuan Chen, Limin Wang, Dahua Lin, Yu Qiao, and Ziwei Liu. VBench: Comprehensive Benchmark Suite for Video Generative Models. In CVPR, 2024.
>
> [9] Xinyuan Chen, Yaohui Wang, Lingjun Zhang, Shaobin Zhuang, Xin Ma, Jiashuo Yu, Yali Wang, Dahua Lin, Yu Qiao, and Ziwei Liu. Seine: Short-to-long video diffusion model for generative transition and prediction. In ICLR, 2024.
>
> [10] Yuwei Guo, Ceyuan Yang, Anyi Rao, Maneesh Agrawala, Dahua Lin, and Bo Dai. SparseCtrl: Adding Sparse Controls to Text-to-Video Diffusion Models. arXiv preprint arXiv:2311.16933, 2023.
>
> [11] Min Zhao, Hongzhou Zhu, Chendong Xiang, Kaiwen Zheng, Chongxuan Li, and Jun Zhu. Identifying and Solving Conditional Image Leakage in Image-to-Video Diffusion Model. In NeurIPS, 2024.

---

> ### Comment · Reviewer_pN4E · 2024-11-27
> **Please Stop Distorting or Ignoring my Reviews**
>
> ***1. Distorting***
>
> You claim that "The transformation starting point of all frames comes from the transformation of the same frame." in my review is a misunderstanding. But I think you are distorting my review.
>
> In your pipeline, z_0 = Bridge Process( Duplicate( ) ). Before the Bridge Process, all information is the same at the frame level. In diffusion models, z_0 = Diffusion Process( Random Sample( ) ). Before the Diffusion Process, all information is different at the frame level. Therefore, Diffusion Models may have more dynamics among frames, which aligns with the phenomenon in this paper. In my reviews, the "transformation" is your Bridge Process (I never say it is a one-step transformation, stop distorting please), and "the same frame" means the input of the Bridge Process. Is that clear? If it is not true, please fix your pipeline figure.
>
> ***2. Ignoring***
>
> Weakness 2: some videos on the website can not be played. It seems that authors still modify the website when the review period starts.
>
> Your response: we will update our project page later and show more cases generated by a FrameBridge model trained longer (i.e. 100k iterations).
>
> If my statement is wrong, please feel free to correct it. Or if you want to skip this weakness, please do not say "Reply to W2". **Are authors allowed to make unlimited changes to the anonymous website after reviewers have started reviewing the manuscript?**
>
> ***3. Video Quality***
>
> In my opinion, it is a reasonable weakness. I have read many generative-related papers in ICLR 25, many reviewers challenge the generated quality even if the method beats the baseline line methods overall. If ACs think reviewers should not do that, please stop all of them. If other submitting papers face this problem, why this article can be a special case?

---

> > ### Author Response · Authors · 2024-11-28
> >
> > We sincerely apologize that we have not addressed your concerns. We value the feedbacks and suggestions provided by all the reviewers and it was not our intention to misinterpret or overlook them. We would truly appreciate the opportunity for a more detailed exchange.
> >
> > **1. Bridge models and dynamic degree**
> >
> > We apologize for not fully understanding the reviewer's concerns in our initial response. We will provide a revised reply about that and hope that it will better address the concerns.
> >
> > As we understand it, the reviewer's concerns lie in that: Each frame in the prior of FrameBridge (i.e., $z_T$ constructed by duplicating the image condition) contains the same information, while diffusion models use a random noise containing different information in each frame as the beginning of sample process. And the reviewer concerns that this will lead to smaller dynamic degree of videos generated by FrameBridge when compared with diffusion I2V models.
> >
> > We would like to show that it is not a weakness of FrameBridge with theoretical analysis and experimental results.
> >
> > **Theory**
> >
> > As is stated by the reviewer, diffusion models use a backward diffusion process to sample the data $z_0$, i.e.:
> > $$z_T \sim \mathcal{N}(0, \sigma_T^2 I), \quad z_0 = Backward\_Diffusion\_Process(z_T) $$
> >
> > Forward diffusion process transforms the data distribution $p_{diff, 0} = p_{data}$ to $p_{diff, T} = \mathcal{N}(0, I)$. As an inversion, the backward diffusion process transforms $\mathcal{N}(0, I)$ back to $p_{data}$. The training of diffusion models is to learn the $Backward\_Diffusion\_Process(\cdot)$ with denoising score matching.
> >
> > For bridge models (we follow the setting of DDBM [1]), there is also a forward bridge process which transforms the data distribution $p_{bridge, 0} = p_{data}$ to $p_{bridge, T}$. Here $p_{bridge, T}$ is a Dirac distribution (i.e., $z_T$ is a deterministic point). This is what leads to the concerns if we understand it correctly.
> >
> > Theoretically, there exists a $Backward\_Bridge\_Process(\cdot)$ such that it transforms $p_{bridge, T - \epsilon}$ back to $p_{bridge, \delta}$, for each $\epsilon, \delta > 0$. Similar to diffusion models, the training of bridge models is to learn the $Backward\_Bridge\_Process(\cdot)$ with denoising bridge score matching. The sampling process of bridge models can be divided into two stages [1]:
> >
> > 1. For a sufficiently small $\epsilon$, we get an approximation of $z_{T - \epsilon}$. As discussed in [1], if we sample with SDE solver, we can approximate $z_{T - \epsilon}$ with $z_{T - \epsilon} \approx z_T$. If we sample with ODE solver, we can approximate $z_{T - \epsilon}$ with an Euler-Maruyama step from $z_T$.
> >
> > 2. We use $Backward\_Bridge\_Process(z_{T - \epsilon})$ to obtain a $z_\delta$ for sufficiently small $\delta$, and we use $z_0 \approx z_{\delta}$ as the result of sampling process.
> >
> > When we use an ODE solver, $Backward\_Bridge\_Process(z_{T - \epsilon})$ can also be written as $Backward\_Bridge\_Process(Euler-Maruyama(z_T, Random\ Noise()))$. When we use an SDE solver, each sampling step of $Backward\_Bridge\_Process(z_T)$ will add Gaussian noise to $z_T$, which will make the identical information of each frame become different. The above analysis does not rely on the choice of $z_T$ and using the duplicated image as $z_T$ will not constrain the dynamic degree of video samples theoretically, and thus we think it is not a weakness of our model.

---

> > ### Author Response · Authors · 2024-11-28
> >
> > **Experimemnts**
> >
> > Furthermore, we adopt reviewer pN4E's suggestions and evaluate the quality of videos generated by FrameBridge on VBench-I2V [2]. We compare it with baseline models (baseline models are all diffusion I2V models) and show the results below:
> >
> > **Table: Scores of VBench-I2V (higher the better)**
> > Model     | I2V Subject Consistency | I2V Background Consistency | Temporal Flickering | Motion Smoothness | Dynamic Degree | Aesthetic Quality | Imaging Quality
> > -------- | -------- | ------- | ----- | -----  | -----  | -----  | -----  |
> > DynamiCrafter | 95.40 | 96.22 | 97.03 | 97.82 | 38.69 | **59.40** | 62.29
> > SEINE [3] | 93.45 | 94.21 | 95.07 | 96.20 | 24.55 | 56.55 | **70.52**
> > SparseCtrl [4] | 88.39 | 92.46 | 91.78 | 94.25 | **81.95** | 49.88 | 69.35
> > ConsistI2V [5]* | 95.82 | 95.95 | 97.56 | 97.38 | 18.62 | 59.00 | 66.92
> > FrameBridge  | **96.24** | **97.25** | **98.01** | **98.51** | 35.77 | 59.38 | 63.28
> >
> > *: We directly use the data from [VBench-I2V Leaderboard](https://huggingface.co/spaces/Vchitect/VBench_Leaderboard).
> >
> > - We admit that the "Dynamic Degree" score of FrameBridge does not outperform all the other diffusion baselines. However, we beg to differ on the point that "all generated results only contain small motions", and it is not an inherent weakness of FrameBridge when compared with diffusion models as shown by the "Dynamic Degree" score in the above table.
> > -  There is usually a trade-off between dynmaic motion and condition alignment [6] for I2V models, and the high dynamic degree scores of some baseline models are at the cost of condition and temporal consistency. FrameBridge can reach a balance demonstrated by the multi-dimensional evaluation on VBench-I2V.
> >
> > **2. About project page**
> >
> > We are sorry that some video samples cannot be played on our website. Now, we have fixed them. As to the website revision, we add figures of our paper on our website on 20-Otc-2024, as we think these may be useful for reviewers to understand our method, and we did not make any changes on our models and methods since the review periord has started. If this operation is inappropriate, we can remove these figures and recover the original version according to the reviewer's suggestion. For high-quality video samples, we are generating them with our new model, i.e., fine-tuning with 100k iterations. We will provide the new results on a new website (if this is preferred) soon.
> >
> > **3. Video Quality**
> >
> > As authors, we do not think our paper should be a special case. Given limited computational resources, we have made our efforts to present useful I2V methods, which achieve stronger experiment results than diffusion-based I2V systems. During this rebuttal process, through fine-tuning with 100k iterations, our results have been much stronger than previous ones (20k iterations), as shown by our results in Table 1 and Table 2 of our new submission.
> >
> > [1] Linqi Zhou, Aaron Lou, Samar Khanna, and Stefano Ermon. Denoising diffusion bridge models. In ICLR, 2024.
> >
> > [2] Ziqi Huang, Yinan He, Jiashuo Yu, Fan Zhang, Chenyang Si, Yuming Jiang, Yuanhan Zhang, Tianxing Wu, Qingyang Jin, Nattapol Chanpaisit, Yaohui Wang, Xinyuan Chen, Limin Wang, Dahua Lin, Yu Qiao, and Ziwei Liu. VBench: Comprehensive Benchmark Suite for Video Generative Models. In CVPR, 2024.
> >
> > [3] Xinyuan Chen, Yaohui Wang, Lingjun Zhang, Shaobin Zhuang, Xin Ma, Jiashuo Yu, Yali Wang, Dahua Lin, Yu Qiao, and Ziwei Liu. Seine: Short-to-long video diffusion model for generative transition and prediction. In ICLR, 2024.
> >
> > [4] Yuwei Guo, Ceyuan Yang, Anyi Rao, Maneesh Agrawala, Dahua Lin, and Bo Dai. SparseCtrl: Adding Sparse Controls to Text-to-Video Diffusion Models. arXiv preprint arXiv:2311.16933, 2023.
> >
> > [5] Weiming Ren, Huan Yang, Ge Zhang, Cong Wei, Xinrun Du, Wenhao Huang, and Wenhu Chen. ConsistI2V: Enhancing Visual Consistency for Image-to-Video Generation. arXiv preprint arXiv:2402.04324, 2024.
> >
> > [6] Min Zhao, Hongzhou Zhu, Chendong Xiang, Kaiwen Zheng, Chongxuan Li, and Jun Zhu. Identifying and Solving Conditional Image Leakage in Image-to-Video Diffusion Model. In NeurIPS, 2024.

---

> > > ### Comment · Reviewer_pN4E · 2024-12-02
> > >
> > > Thank you for the reply, I think there is no misunderstanding now. Although the author put a lot of effort into the rebuttal, I still feel that this paper is not ready for ICLR acceptance.
> > >
> > > Even for the new video results of 100k (submitted during the rebuttal period), video quality is still unsatisfactory. An obvious flaw is that the high-frequency details in the input image disappear in the generated result. I think the authors may also be aware of this problem, but due to time constraints, they are unable to resolve it and can only upload the current version. I do not deny the potential of the framework proposed by the author, but the author's experimental results cannot prove its superiority.
> > >
> > > At the same time, some operations suspected of violating regulations are secondary reasons for rejecting this paper. The quality of this paper is not amazing enough for me to ignore that behavior.

---

> > > > ### Author Response · Authors · 2024-12-02
> > > >
> > > > Dear Reviewer pN4E,
> > > >
> > > > We deeply appreciate your time and effort in reviewing our responses and engaging in the discussion with us. It is encouraging that the potential of our method has been acknowledged and that misunderstandings can be eliminated through discussion. Thanks for your advice and pointing out the aspects of our manuscript and experiments that need improvement, and we will make an effort to improve our work in the future.

---

### Official Review · Reviewer_yg8s · 2024-10-30

**Soundness:** 3
**Presentation:** 3
**Contribution:** 3
**Rating:** 5
**Confidence:** 5

**Summary:**

This paper proposes a novel image-to-video (I2V) generation framework called FrameBridge, which models the frame-to-frames synthesis process with a data-to-data generative framework. Unlike traditional diffusion-based I2V methods, FrameBridge takes the given static image as the prior of the video target, establishing a bridge model between them. This approach reduces the burden of generative models and improves synthesis quality. The authors also propose two techniques, SNR-Aligned Fine-tuning (SAF) and neural prior, to further enhance the performance of FrameBridge.

**Strengths:**

1.	FrameBridge introduces a new data-to-data generation framework for I2V synthesis, which is different from traditional noise-to-data diffusion-based methods.
2.	By taking the given static image as the prior of the video target, FrameBridge reduces the burden of generative models and improves synthesis quality.
3.	The proposed SAF technique enables seamless knowledge transfer between pre-trained diffusion models and FrameBridge, improving fine-tuning efficiency.

**Weaknesses:**

1.	My primary concern about this paper is the experiment. I acknowledge that the bridge model may be a promising method for image-to-video diffusion models, and this paper has made an effort to fully utilize input image information. However, this paper only provides baselines for SEINE and DynamiCrafter, other works such as Animate Anyone [1] and Sparsectrl [2] have attempted to introduce an additional appearance encoding and fusion module to support more refined conditional image information utilization, which is an advantage over these methods. Therefore, this paper needs to provide a more detailed discussion. Moreover, ConsistI2V [3] designed FrameInit, which incorporates image priors into the initial noise, breaking the noise-to-data pattern and leading to better results. This paper does not provide a comprehensive comparison with other state-of-the-art I2V generation methods.
2.	Regarding evaluation metrics, FVD and IS are no longer reliable metrics. In fact, there are many benchmarks focused on video generation, such as T2V-CompBench [4] and VBench [5], which provide reliable evaluation metrics. The authors should use the most advanced benchmarks. Furthermore, this paper does not provide user studies.

[1] Animate Anyone: Consistent and Controllable Image-to-Video Synthesis for Character Animation, Hu et al., CVPR 2024
[2] Sparsectrl: Adding sparse controls to text-to-video diffusion models, Guo et al., ECCV 2024
[3] ConsistI2V: Enhancing Visual Consistency for Image-to-Video Generation, Re et al., TMLR 2024
[4] T2V-CompBench: A Comprehensive Benchmark for Compositional Text-to-video Generationrk, Sun et al., arXiv 2024
[5] VBench: Comprehensive Benchmark Suite for Video Generative Models, Huang et al., CVPR 2024

**Questions:**

Based on the weaknesses mentioned, this paper requires further revisions, therefore, I currently give it a score of 'marginally below the acceptance threshold'.

---

> ### Comment · Reviewer_yg8s · 2024-11-26
> **Do the authors still plan to submit a rebuttal?**
>
> Do the authors still plan to submit a rebuttal?

---

> > ### Author Response · Authors · 2024-11-26
> >
> > Yes, we are now carefully prepare the materials and will submit a rebuttal soon. As our computational resources are very limited, it takes us some time to obtain the experimental results. We are glad to engage in a detailed and thorough discussion later.

---

> ### Author Response · Authors · 2024-11-27
>
> We appreciate the reviewer for acknowledging the novelty and the potential of our approach. To address the reviewer's concerns, we provide more experimental results and discussions about related works below:
>
> **Reply to W1: Comparisons and discussions for more baselines**
>
> **1. Comparisons with more baselines:**
>
> We add SparseCtrl [1] and ConsistI2V [2] as baselines and provide the evaluation results of zero-shot metrics on MSR-VTT and UCF-101. We also adopt the suggestions of other reviewers and include the results of some other classic I2V diffusion models (SVD [3] and I2VGen-XL [4]).
>
> **Table: Zero-shot metrics on MSR-VTT**
>
> Model     | FVD $\downarrow$ | CLIPSIM $\uparrow$ | PIC $\uparrow$
> -------- | -------- | ------- | -----
> SparseCtrl | 331.44| 0.2245 | 0.4382
> ConsistI2V | 105.35 | **0.2250** | 0.7530
> I2VGen-XL* | 289.10 | - | 0.5352
> SVD** |  114.61 | - | **0.8547**
> DynamiCrafter (released checkpoint [7]) | 190.04 | 0.2245 | 0.6209
> SEINE | 176.96 | **0.2250** | 0.4217
> FrameBridge (trained 100k iters)  | **77.65** | **0.2250** | 0.7477
>
> **Table: Zero-shot metrics on UCF-101**
>
> Model     | FVD $\downarrow$ | IS $\uparrow$ | PIC $\uparrow$
> -------- | -------- | ------- | -----
> SparseCtrl | 722.43 | 19.45 | 0.4818
> ConsistI2V | **182.72** | 39.30 | 0.7653
> I2VGen-XL* | 571.11 | - | 0.5313
> SVD** | 234.88 | - | **0.8527**
> DynamiCrafter (releaesd checkpoint) | 484.66 | 29.46 | 0.6266
> SEINE | 461.10 | 22.32 | 0.6665
> FrameBridge (trained 100k iters)  | 213.44 | **43.36** | 0.7633
>
> *: We directly use the data from [7].
>
> **: Since the I2V model of SVD generate videos with 14 frames (all the other models in the above tables generate videos with 16 frames), the PIC metric (average of similarity between each frame and the image condition) of SVD may be overestimated as the image condition is used as the first frame.
>
> Besides, we provide some additional clarification below:
>
> - We do not add Animate Anyone [5] as it is an image animation method designed for character animation and requires additional conditions (i.e., pose sequence), which is a different setting from FrameBridge and other baseline models. Nevertheless, we will discuss this work later.
>
> - For ConsistI2V, we use the officially released model weights. Although their model is trained with much more budget than our FrameBridge model, the video quality of FrameBridge is comparable to theirs.
>
> **Table: Training budget and dataset size for different models**
> Model | Batch Size | Training Iterations | Dataset Size
> --- | --- | --- | ---
> ConsistI2V | 192 | 170k | 10M (WebVid-10M)
> DynamiCrafter | 64 | 100k | 10M (WebVid-10M)
> FrameBridge | 64 | 100k | 2M (WebVid-2M)
>
> - We will provide the comprehensive evaluation results on VBench [6] in the following part.
>
> **2. Discussions about previous works**
>
> - **I2V diffusion models with additional fusion module:** Some previous works [1, 5, 8, 9] have explored to enhance T2I or T2V diffusion models with additional modules to achieve controllable video generation. It is a constructive methodology as how to integrate conditions into the denoising network effectively is crucial for I2V and other controllable video generation tasks. In our work, we aim at replacing the conditional diffusion process with a frames-to-frames bridge process, and trains a bridge denoising network to sample videos. Our FrameBridge is parallel to these works, and we can freely choose the structure of the denoising network (e.g., in our experiments, we use the same network structure as DynamiCrafter [7] which also utilizes a lightweight projection module to process the image condition).
>
> - **Noise manipulation for video diffusion models:** Several works have explored to improve the uninformative prior distribution of diffusion models. PYoCo [10] proposes to use correlated noise for each frame in both training and inference. ConsistI2V [2], FreeInit [11], and CIL [12] present training-free strategies to better align the training and inference distribution of diffusion prior. These strategies focus on improving the noise distribution to enhance the quality of synthesized videos, while they still suffer the restriction of noise-to-data diffusion framework, which may limit their endeavor to utilize the entire information (e.g., both large-scale features and fine-grained details) contained in the given image. In contrast, we propose a data-to-data framework and utilize deterministic prior rather than Gaussian noise, allowing us to leverage the clean input image as prior information.
>
> We appreciate the reviewer's advice and will add these discussions in the final version of our paper later.

---

> ### Author Response · Authors · 2024-11-27
>
> **Reply to W2: Evaluation of I2V models**
>
> As suggested by reviewer yg8s, we evaluate baselines and FrameBridge on an advanced I2V benchmark, i.e. VBench-I2V[6], and shows the results below:
>
> **Table: Scores of VBench-I2V (higher the better)**
> Model     | I2V Subject Consistency | I2V Background Consistency | Temporal Flickering | Motion Smoothness | Dynamic Degree | Aesthetic Quality | Imaging Quality
> -------- | -------- | ------- | ----- | -----  | -----  | -----  | -----  |
> DynamiCrafter | 95.40 | 96.22 | 97.03 | 97.82 | 38.69 | **59.40** | 62.29
> SEINE | 93.45 | 94.21 | 95.07 | 96.20 | 24.55 | 56.55 | **70.52**
> SparseCtrl | 88.39 | 92.46 | 91.78 | 94.25 | **81.95** | 49.88 | 69.35
> FrameBridge  | **96.24** | **97.25** | **98.01** | **98.51** | 35.77 | 59.38 | 63.28
>
> - The above table demonstrates the high level overall quality of videos generated by FrameBridge, and FrameBridge outperforms all the other baseline models across most dimensions of video quality (especially for those related to I2V Consistency). Considering the training budget and dataset size, FrameBridge is a promising method for I2V generation.
>
> - About dynamic degree: There is usually a trade-off between dynmaic motion and condition alignment [13] for I2V models, and the high dynamic degree scores of some baseline models are at the cost of condition and temporal consistency. FrameBridge can reach a balance demonstrated by the multi-dimensional evaluation on VBench-I2V. Moreover, the motion dynamic of FrameBridge is not low compared with other models on
> [VBench-I2V leaderboard](https://huggingface.co/spaces/Vchitect/VBench_Leaderboard).
>
> **Table: Scores related to motion**
> Model     | Dynamic Degree | Temporal Flickering | Motion Smoothness
> -------- | -------- | ---- | --- |
> FrameBrdige | 35.77 | 98.01 | 98.51
> SEINE [7]-512 $\times$ 320* | 34.31 | 96.72 | 96.68
> SEINE-512 $\times$ 512* | 27.07 | 97.31 | 97.12
> ConsistI2V [8]* | 18.62 | 97.56 | 97.38
>
> *: We directly use the evaluation results provided by [VBench-I2V leaderboard](https://huggingface.co/spaces/Vchitect/VBench_Leaderboard).

---

> ### Author Response · Authors · 2024-11-27
>
> Due to the constraints of time and computational resources, we are unable to provide the results of user study currently. We will add it in the subsequent discussion period.
>
> We hope the above experimental results and discussions can address the reviewer's concerns. We would be glad to have a further discussion.
>
> [1] Yuwei Guo, Ceyuan Yang, Anyi Rao, Maneesh Agrawala, Dahua Lin, and Bo Dai. SparseCtrl: Adding Sparse Controls to Text-to-Video Diffusion Models. arXiv preprint arXiv:2311.16933, 2023.
>
> [2] Weiming Ren, Huan Yang, Ge Zhang, Cong Wei, Xinrun Du, Wenhao Huang, and Wenhu Chen. ConsistI2V: Enhancing Visual Consistency for Image-to-Video Generation. arXiv preprint arXiv:2402.04324, 2024.
>
> [3] Andreas Blattmann, Tim Dockhorn, Sumith Kulal, Daniel Mendelevitch, Maciej Kilian, Dominik Lorenz, Yam Levi, Zion English, Vikram Voleti, Adam Letts, Varun Jampani, and Robin Rombach. Stable Video Diffusion: Scaling Latent Video Diffusion Models to Large Datasets. arXiv preprint arXiv:2311.15127, 2023.
>
> [4] Shiwei Zhang, Jiayu Wang, Yingya Zhang, Kang Zhao, Hangjie Yuan, Zhiwu Qin, Xiang Wang, Deli Zhao, and Jingren Zhou. I2VGen-XL: High-Quality Image-to-Video Synthesis via Cascaded Diffusion Models. arXiv preprint arXiv:2311.04145, 2023.
>
> [5] Li Hu, Xin Gao, Peng Zhang, Ke Sun, Bang Zhang, and Liefeng Bo. Animate Anyone: Consistent and Controllable Image-to-Video Synthesis for Character Animation. arXiv preprint arXiv:2311.17117, 2023.
>
> [6] Ziqi Huang, Yinan He, Jiashuo Yu, Fan Zhang, Chenyang Si, Yuming Jiang, Yuanhan Zhang, Tianxing Wu, Qingyang Jin, Nattapol Chanpaisit, Yaohui Wang, Xinyuan Chen, Limin Wang, Dahua Lin, Yu Qiao, and Ziwei Liu. VBench: Comprehensive Benchmark Suite for Video Generative Models. In CVPR, 2024.
>
> [7] Jinbo Xing, Menghan Xia, Yong Zhang, Haoxin Chen, Wangbo Yu, Hanyuan Liu, Gongye Liu, Xintao Wang, Ying Shan, and Tien-Tsin Wong. DynamiCrafter: Animating Open-Domain Images with Video Diffusion Priors. In ECCV, 2024.
>
> [8] Xiang Wang, Hangjie Yuan, Shiwei Zhang, Dayou Chen, Jiuniu Wang, Yingya Zhang, Yujun Shen, Deli Zhao, and Jingren Zhou. Videocomposer: Compositional video synthesis with motion controllability. In NeurIPS, 2024.
>
> [9] Yuwei Guo, Ceyuan Yang, Anyi Rao, Zhengyang Liang, Yaohui Wang, Yu Qiao, Maneesh Agrawala, Dahua Lin, and Bo Dai. Animatediff: Animate your personalized text-to-image diffusion models without specific tuning. arXiv preprint arXiv:2307.04725, 2023.
>
> [10] Songwei Ge, Seungjun Nah, Guilin Liu, Tyler Poon, Andrew Tao, Bryan Catanzaro, David Jacobs, Jia-Bin Huang, Ming-Yu Liu, and Yogesh Balaji. Preserve your own correlation: A noise prior for video diffusion models. In ICCV, 2023.
>
> [11] Tianxing Wu, Chenyang Si, Yuming Jiang, Ziqi Huang, and Ziwei Liu. Freeinit: Bridging initialization gap in video diffusion models. arXiv preprint arXiv:2312.07537, 2023.
>
> [12] Min Zhao, Hongzhou Zhu, Chendong Xiang, Kaiwen Zheng, Chongxuan Li, and Jun Zhu. Identifying and Solving Conditional Image Leakage in Image-to-Video Diffusion Model. In NeurIPS, 2024.

---

> > ### Comment · Reviewer_yg8s · 2024-12-02
> > **Official Comment by Reviewer yg8s**
> >
> > Thank you for the author's rebuttal and further experiments and discussions. I have read all the comments from the reviewers and the corresponding responses. Overall, I agree with reviewer pN4E's concerns about video quality. The authors do not have sufficient evidence to demonstrate the superiority of the proposed method. I appreciate the additional experimental results provided by the authors on VBench, but I still have doubts about fair comparisons. As it is well known, the quality of visual content is difficult to fully measure quantitatively, so qualitative comparisons and user studies are necessary. However, user studies are missing. Additionally, qualitative comparisons in this paper only provide two examples, which I believe is insufficient. Furthermore, DynamiCrafter, which has the best performance on VBench, seems to perform unsatisfactorily in this paper. I personally ran the official implementation of DynamiCrafter (https://github.com/Doubiiu/DynamiCrafter) and tried the two examples provided on the project homepage: (1) "One girl is waving hand, one girl, solo, sexy pose, pensive woman, voluminous dress, intricate lace, embroidered gloves, feathered hat, curled hairdo, pale skin" and (2) "sun rising, mountain, path, masterpiece." I obtained results far better than those provided by the authors, with no unnecessary artifacts. Therefore, I have doubts about the fairness of the experiments. Overall, I still believe that this manuscript is not ready for publication at ICLR.

---

> > > ### Author Response · Authors · 2024-12-02
> > >
> > > Dear Reviewer yg8s,
> > >
> > > We are deeply grateful for your time and effort to review our work. We sincerely appreciate your valuable suggestions and for pointing out the aspects of our manuscript and experiments that need improvement, and will continue to improve our method and models in the future.

---

### Official Review · Reviewer_mfhQ · 2024-11-03

**Soundness:** 2
**Presentation:** 3
**Contribution:** 2
**Rating:** 3
**Confidence:** 4

**Summary:**

The paper introduces FrameBridge, which formulates I2V synthesis as a frames-to-frames generation task rather than a conditional noise-to-frames generation. This approach allows the model to concentrate more effectively on learning image animation, enhancing both the temporal and appearance coherence of the generated video. Furthermore, to better utilize the priors from pre-trained T2V models, SNR Aligned Fine-tuning (SAF) and a neural prior are proposed to promote fine-tuning efficiency and improve the quality of generated videos. Extensive experiments demonstrate the effectiveness of FrameBridge in image-to-video generation tasks.

**Strengths:**

S1: Unlike previous diffusion-based methods, FrameBridge formulates I2V synthesis as an interesting frames-to-frames generation task rather than a conditional noise-to-frames generation.

S2: The paper presents good qualitative and quantitative results.

**Weaknesses:**

W1: In Table 1, some important baselines are missing, such as SVD [r1] and I2VGen-XL [r2], both of which are strong baselines and would be valuable for comparison. Additionally, I noticed in line 473 that DynamiCrafter was reproduced in this paper. I’m curious why the original weights couldn’t be used for testing, as the videos generated by DynamiCrafter in Figure 5 and on the project page seem to exhibit significant inconsistencies. This seems unusual, given that DynamiCrafter claims to rank first on the I2V benchmark of VBench [r3]. Clarifying the reproduction process and discussing potential discrepancies would be helpful.

[r1] Blattmann, Andreas, et al. "Stable video diffusion: Scaling latent video diffusion models to large datasets." arXiv preprint arXiv:2311.15127 (2023).
[r2] Zhang, Shiwei, et al. "I2vgen-xl: High-quality image-to-video synthesis via cascaded diffusion models." arXiv preprint arXiv:2311.04145 (2023).
[r3] Huang, Ziqi, et al. "Vbench: Comprehensive benchmark suite for video generative models." Proceedings of the IEEE/CVF Conference on Computer Vision and Pattern Recognition. 2024.

W2: In Figure 5 and on the project page, most of the generated videos seem to exhibit only minor movements, which raises a concern that the proposed method may tend to produce videos with limited motion. Could the authors provide examples of the method generating videos with larger-scale actions, such as camera movements or vehicles in motion? Alternatively, discussing any limitations in generating more dynamic scenes would be valuable.

W3: In line 470 of the paper, it is noted that all video synthesis methods sample 250 steps, which is quite time-consuming. Since most models achieve good results with only 50 steps, could the authors clarify the justification for using 250 steps? Additionally, it would be helpful if the authors could provide a comparison of quality vs. number of steps, along with the inference time for different step counts.

W4: In Table 2, FrameBridge-NP is mentioned. Could the authors clarify whether the default FrameBridge includes a neural prior? Additionally, it would be helpful to specify if NP is used only in the context of training from scratch or if it is a default setting. If NP is only applicable or beneficial in specific training scenarios, clarifying these conditions would be useful.

W5: Table 2 should include experimental results of classic I2V network architectures, such as Dynamicrafter and I2VGEN-XL. Furthermore, I think it may be better to conduct ablation studies based on the setting in Table 1, as suggested in the Dynamicrafter paper, since generalizability is crucial for image-to-video generation models.

**Questions:**

I hope the authors can address the concerns raised in the "Weaknesses" section.

---

> ### Author Response · Authors · 2024-11-26
>
> We thank the reviewer for their constructive and practicable suggestions to improve the experiment part of this paper. In response, we have included additional results to address the reviewer’s concerns:
>
> **Reply to W1: Evaluation of baselines**
>
> **DynamiCrafter:** In the first version of our paper, we do not use the original weights of DynamiCrafter for evaluation as there are some differences between the setting in our work and DynamiCrafter [1].
>  - **Training and sampling settings:** Following previous works of Image-to-Video (I2V) generation [2, 3, 4], the target of FrameBridge is to generate videos with condition image as **the first frame**. In contrast, DynamiCrafter adopts a different setting and use **a randomly selected frame** from the whole video as image condition and there may be discrepency between the beginning of generated videos and image conditions.
>  - **Training budget:** For the original weights of DynamiCrafter, the I2V diffusion model is fine-tuned from a T2V diffusion model for 100k iterations, and FrameBridge is fine-tuned from the same T2V diffusion model for 20k iterations because of our very limited computational resources.
>  Therefore, to compare FrameBridge and DynamiCrafter under the same experiment setting and budget, we reproduce DynamiCrafter at 256 $\times$ 256 resolution for 20k iterations with their official implementation, except that image conditions are sampled as first frames of video data.
>
> In this rebuttal, to address the reviewer's concerns, we evaluate the original weights of DynamiCrafter and compare it with a FrameBridge model trained longer (i.e. 100k iterations):
>
> **Table: Metrics on MSR-VTT**
> Model     | FVD $\downarrow$ | CLIPSIM $\uparrow$ | PIC $\uparrow$
> -------- | -------- | ------- | -----
> DynamiCrafter (original weights) | 190.04 | 0.2245 | 0.6209
> FrameBridge (20k iters)  | 83.33 | **0.2249** | 0.7185
> FrameBridge (100k iters)  | **77.65** | **0.2249** | **0.7477**
>
> Moreover, as suggested by Reviewer mfhQ, we also compare DynamiCrafter (original weights at 256 $\times$ 256) and FrameBridge (100k iters) on VBench-I2V [5]. (The model ranked first on VBench-I2V is the 576 $\times$ 1024 version of DynamiCrafter)
>
> **Table: Scores of VBench-I2V (higher the better)**
> Model     | I2V Subject Consistency | I2V Background Consistency | Temporal Flickering | Motion Smoothness | Dynamic Degree | Aesthetic Quality | Imaging Quality
> -------- | -------- | ------- | ----- | -----  | -----  | -----  | -----  |
> DynamiCrafter (original weights) | 95.40 | 96.22 | 97.03 | 97.82 | 38.69 | 59.40 | 62.29
> FrameBridge (100k iters)  | 96.24 | 97.25 | 98.01 | 98.51 | 35.77 | 59.38 | 63.28
>
> FrameBridge outperforms DynamiCrafter-256 at different dimensions except dynamic degree. However, the higher score may be caused by more flickering and less temporal consistency (which can be justified by the temporal flickering score and motion smoothness score). We will have a further discussion about the dynamic degree of FrameBridge latter.
>
> **Other baselines:** We adopt the reviewer's suggestions and add SVD[2], I2VGen-XL[6] as baselines. Below we provide the evaluation of sample results with 50 sample steps on MSR-VTT (we will add both results on MSR-VTT and UCF-101 in the final version of our paper):
>
> **Table: Metrics on MSR-VTT (50 sample steps)**
> Model     | FVD $\downarrow$ | CLIPSIM $\uparrow$ | PIC $\uparrow$
> -------- | -------- | ------- | -----
> I2VGen-XL* | 289.10 | - | 0.5352
> SVD** | 114.61 |  - | **0.8547**
> DynamiCrafter (original weights) |191.76 | 0.2245 | 0.6131
> DynamiCrafter (reported in [1]) | 234.66 | - | 0.5803
> SEINE | 416.98 | **0.2250** | 0.6431
> FrameBridge (20k iters)  | 99.39 | **0.2250** | 0.6963
> FrameBridge (100k iters)  | **95.20** | **0.2250** | 0.7142
>
> *: Considering the resolution of released model, the inference of I2VGen-XL requires much computational resources, and there is no official evaluation result on MSR-VTT [6], we use the result reported in [1].
>
> **: The I2V model of SVD only conditions on the input image and does not use text prompts, so we don't evaluate the CLIP similarity between video frames and text prompts. Besides, SVD generates 14 frames and all the other models in the table generate 16 frames. This may lead to higher average similarity between generated frames and the condition image (i.e. higher PIC) of SVD as all models use the condition image as the first frame in generated videos.

---

> ### Author Response · Authors · 2024-11-26
>
> **Reply to W2: Motion scale of generated videos**
>
> **Samples with larger movements:** We will provide more samples with larger movements on our project page.
>
> **Metric of dynamic degree:** We use VBench-I2V [5] to evaluate the dynamic degree of generated samples. As mentioned above, although the dynamic degree of DynamiCrafter is higher than FrameBridge, it is at the cost of temporal consistency and motion smoothness. Moreover, the motion dynamic of FrameBridge is not low compared with other models on
> [VBench-I2V leaderboard](https://huggingface.co/spaces/Vchitect/VBench_Leaderboard).
>
> **Table: Scores of VBench-I2V**
> Model     | Dynamic Motion | Temporal Flickering | Motion Smoothness
> -------- | -------- | ---- | --- |
> DynamiCrafter-256 (original weights) | 38.69 | 97.03 | 97.82
> FrameBrdige (100k iters) | 35.77 | 98.01 | 98.51
> SEINE [7]-512 $\times$ 320* | 34.31 | 96.72 | 96.68
> SEINE-512 $\times$ 512* | 27.07 | 97.31 | 97.12
> Consist-I2V [4]* | 18.62 | 97.56 | 97.38
>
> *: We directly use the data provided by [VBench-I2V leaderboard](https://huggingface.co/spaces/Vchitect/VBench_Leaderboard).
>
> We would like to clarify that the motion scale of generated videos are influenced by several experimental factors independent of models and algorithms. For example, training dataset, text prompts and fps condition used when sampling, and data preprocess strategy  (e.g., frame stride of training data) all play a role in determining the motion scale of videos generated by I2V models. Meanwhile, there is usually a trade-off between dynmaic motion and image alignment [8] for I2V models, and our FrameBridge aims at reaching a balance demonstrated by the multi-dimensional evaluation on VBench-I2V.
>
>
> **Reply to W3: Sampling steps**
>
> **Justification for using 250 steps:** Using more timesteps will guarantee the covergence of the solution for SDEs/ODEs and reduce numerical errors caused by the sampler. In our experiments, we focus on comparing the I2V generation capabilities between bridge-based I2V models and their diffusion counterparts, so we use a sufficiently large timestep to
> fully demonstrate the generative capability of different models and reduce the impact of numerical errors.
>
> **Quality vs. Number of timesteps:** We evaluate the impact of sampling timesteps for quality on MSR-VTT datasets as below:
> (We will add figures to illustrate that in our final version of paper later)
>
> **Table: FVD on MSR-VTT (lower the better)**
> Model     | 250 timesteps | 100 timesteps | 50 timesteps | 40 timesteps | 20 timesteps
> -------- | -------- | ------- | ----- | --- | ---
> DynamiCrafter (original weights) | 190.04 | 190.63 | 191.76 | 194.54 | 191.93
> SEINE (original weights) | 176.96 | 244.90 | 416.98 | 525.84 | 1124.12
> FrameBridge (100k iters)  | **77.65** | **82.65** | **95.20** | **95.81** | **113.04**
>
> FrameBridge has lower FVD (better video quality) compared with its diffusion counterparts using different sampling timesteps.
>
> **Table: PIC on MSR-VTT (higher the better)**
> Model     | 250 timesteps | 100 timesteps | 50 timesteps | 40 timesteps | 20 timesteps
> -------- | -------- | ------- | ----- | --- | ---
> DynamiCrafter (original weights) | 0.6209 | 0.6210 | 0.6131 | 0.6116 | 0.5946
> SEINE (original weights) | 0.4217 | 0.6848 | 0.6431 | 0.6150 | 0.4912
> FrameBridge (100k iters)  | **0.7477** | **0.7328** | **0.7142** | **0.7115** | **0.6900**
>
> As shown, for FrameBridge, the similarity between generated video frames and image conditions  is higher than that of diffusion-based I2V models across different sampling timesteps. Furthermore, the similarity remains relatively stable as the sampling timesteps are reduced, whereas the similarity of diffusion models decreases rapidly.
>
> The wall-clock time required for sampling is directly proportional to sampling timesteps. FrameBridge and DynamiCrafter use networks with the same structure, so their sampling speeds are identical when using the same number of timesteps. We will compute the time required per step for SEINE and compare it with that of FrameBridge/DynamiCrafter later in the final version of our paper.

---

> ### Author Response · Authors · 2024-11-26
>
> **Reply to W4: Clarification of neural prior**
>
> - The default FrameBridge does not include a neural prior. Neural prior is a technique to
> further enhance FrameBridge by replacing the prior of bridge models with the results of another one-step regression model.
>
> - Theoretically, the prior of bridge models can be chosen freely and replicating the first frame (i.e., image condition) is a convenient and effective practice justified by our experiments. The goal of experiments for FrameBridge-NP training from scratch on UCF-101 is to demonstrate that utilizing one-step regression neural prior (although it is a relatively vague and over-smooth intermediate result) is a useful method to improve the performance of default FrameBridge. Due to the limitation of computational resources, we can only fine-tune T2V diffusion models to I2V bridge models with the proposed SAF technique and are unable to train the neural prior model from scratch on WebVid dataset.
>
> **Reply to W5: Clarification of experimental setups**
>
> We want to clarify that the experimental setups in Table 1 and Table 2 are completely different.
>
> All the baseline models and FrameBridge models in Table 1 are **fine-tuned from pre-trained diffusion models**, trained on **large dataset (WebVid2M or WebVid10M)** consisting of **text-video pairs**.
>
> All the baseline models and FrameBridge models in Table 2 are **trained-from scratch** on **relatively small-scale dataset (UCF-101)** consisting of **class-labeled videos**. So, models like DynamiCrafter, I2VGen-XL can not be directly applied to this setting.
>
> **Ablation studies for Table 1:** Part of the original Table 1 can be seen as the results of an ablation study.
> - **Results of FrameBridge (w/o SAF) vs. FrameBridge (w/SAF)** show that FrameBridge with SAF technique performs better than those without SAF.
> - **Results of FrameBridge (w/o SAF) vs. DynamiCrafter** show that bridge process is more suitable for I2V generation than diffusion process (as the two models have the same network structure and are trained with the same budget).
>
> We hope these results and clarifications will address your concerns, and we are willing to have a further discussion.
>
> [1] Jinbo Xing, Menghan Xia, Yong Zhang, Haoxin Chen, Wangbo Yu, Hanyuan Liu, Gongye Liu, Xintao Wang, Ying Shan, and Tien-Tsin Wong. DynamiCrafter: Animating Open-Domain Images with Video Diffusion Priors. In ECCV, 2024.
>
> [2] Andreas Blattmann, Tim Dockhorn, Sumith Kulal, Daniel Mendelevitch, Maciej Kilian, Dominik Lorenz, Yam Levi, Zion English, Vikram Voleti, Adam Letts, Varun Jampani, and Robin Rombach. Stable Video Diffusion: Scaling Latent Video Diffusion Models to Large Datasets. arXiv preprint arXiv:2311.15127, 2023.
>
> [3] Xin Ma, Yaohui Wang, Gengyun Jia, Xinyuan Chen, Yuan-Fang Li, Cunjian Chen, and Yu Qiao. Cinemo: Consistent and Controllable Image Animation with Motion Diffusion Models. arXiv preprint arXiv:2407.15642, 2024.
>
> [4] Weiming Ren, Huan Yang, Ge Zhang, Cong Wei, Xinrun Du, Wenhao Huang, and Wenhu Chen. ConsistI2V: Enhancing Visual Consistency for Image-to-Video Generation. arXiv preprint arXiv:2402.04324, 2024.
>
> [5] Ziqi Huang, Yinan He, Jiashuo Yu, Fan Zhang, Chenyang Si, Yuming Jiang, Yuanhan Zhang, Tianxing Wu, Qingyang Jin, Nattapol Chanpaisit, Yaohui Wang, Xinyuan Chen, Limin Wang, Dahua Lin, Yu Qiao, and Ziwei Liu. VBench: Comprehensive Benchmark Suite for Video Generative Models. In CVPR, 2024.
>
> [6] Shiwei Zhang, Jiayu Wang, Yingya Zhang, Kang Zhao, Hangjie Yuan, Zhiwu Qin, Xiang Wang, Deli Zhao, and Jingren Zhou. I2VGen-XL: High-Quality Image-to-Video Synthesis via Cascaded Diffusion Models. arXiv preprint arXiv:2311.04145, 2024.
>
> [7] Xinyuan Chen, Yaohui Wang, Lingjun Zhang, Shaobin Zhuang, Xin Ma, Jiashuo Yu, Yali Wang, Dahua Lin, Yu Qiao, and Ziwei Liu. SEINE: Short-to-Long Video Diffusion Model for Generative Transition and Prediction. In ICLR, 2024.
>
> [8] Min Zhao, Hongzhou Zhu, Chendong Xiang, Kaiwen Zheng, Chongxuan Li, and Jun Zhu. Identifying and Solving Conditional Image Leakage in Image-to-Video Diffusion Model. In NeurIPS, 2024.

---

> > ### Comment · Reviewer_mfhQ · 2024-12-01
> >
> > Thanks for additional experimental results and discussion in the author responses. However, I totally agree with Reviewer pN4E for the unconvincing video quality compared to state-of-the-art methods. Moreover, in the comparison on VBench-I2V, this work fails to outperform DynamiCrafter on Dynamic Degree and Aesthetic Quality. Actually, the decrease of dynamic degress commonly leads to better Motion Smoothness and Temporal Consistency even for the same method. So I still have the concern on such unconvincing comparison on VBench-I2V with marginal performance improvements. On the updated demo page, the increase of dynamic degress also results in more Temporal Flickering.

---

> > > ### Author Response · Authors · 2024-12-02
> > >
> > > Dear Reviewer mfhQ,
> > >
> > > We greatly appreciate your time and effort in reviewing our response. Thanks for your valuable suggestions to help us improve the experimental results. We are sorry that we do not address your concerns, and we will continue to improve our method and models in the future.

---

### Author Response · Authors · 2024-12-01
**Updated information of FrameBridge**

Dear Reviewers:

We would like to thank all the reviewers' efforts and insightful feedback to help us improve our work. We are sorry for the delay in providing our rebuttal content and improved experiment results. If it is possible, could you please kindly review our revised manuscript and the supplementary demo page (https://framebridgei2v-hq.github.io/). We greatly appreciate your help and response.

### Paper Revisions

We summarize the revisions of our manuscript below:

**More details and discussion of FrameBridge methods**:
1. Discuss about more diffusion-based I2V methods and the alignment of diffusion noise schedules in the Related Works (Section 2), and add more detailed discussions in Appendix B.

2. Discuss the dynamic degree of FrameBridge in Appendix D.1.

3. Use pseudo code to show the training and sampling algorithm in Appendix E.

**Improved Experiment results**:
1. Add more baselines (I2VGen-XL, SVD, ConsistI2V, SparseCtrl) in Table 1 (zero-shot metrics on UCF-101 and MSR-VTT).

2. Compare FrameBridge with other baseline models on VBench-I2V (Table 2 in Section 5.2).

3. Add a new ablation setting to validate the effectiveness of neural prior (Table 5 in Section 5.4).

4. Add evaluation results of content-debiased FVD in certain experimental settings in Appendix D.2.

5. Demonstrate the training and sampling efficiency of FrameBridge in Appendix D.3 and Appendix D.4.


### Demo Page

We kindly recommend reviewers to check our new [demo page](https://framebridgei2v-hq.github.io/) which consists of two sections. In the first section, we provide several video samples of a FrameBridge model trained longer (i.e., FrameBridge-100k). In the second section, we provide some videos with more evident motion, which we hope could address the concerns about dynamic degree of our model.

We would greatly appreciate it if the reviewers could kindly check our updated materials and provide us with an opportunity for further discussion.

---

### Meta-Review · Area_Chair_yKYC · 2024-12-04

**Metareview:**

All reviewers agree to reject the paper. Reviewers mainly complain about the result quality and unfair comparisons to SOTA methods, which could not be addressed in the rebuttal. The authors are encouraged to address the concerns and submit elsewhere.

**Additional Comments On Reviewer Discussion:**

The main concerns raised by several reviewers such as result quality could not be resolved in the rebuttal.

---

### Decision · Program_Chairs · 2025-01-22

Reject